# Different promoter affinities account for specificity in MYC-dependent gene regulation

**Francesca Lorenzin[1], Uwe Benary[2], Apoorva Baluapuri[1], Susanne Walz[3,4], Lisa Anna Jung[1,5], Björn von Eyss[1†], Caroline Kisker[5], Jana Wolf[2], Martin Eilers[1,4], Elmar Wolf[1]\***

[1]Department of Biochemistry and Molecular Biology, Biocenter, University of Würzburg, Würzburg, Germany; [2]Group Mathematical Modeling of Cellular Processes, Max-Delbrück-Center for Molecular Medicine, Berlin, Germany; [3]Core Unit Bioinformatics, Biocenter, University of Würzburg, Würzburg, Germany; [4]Comprehensive Cancer Center Mainfranken, University of Würzburg, Würzburg, Germany; [5]Rudolf-Virchow-Center for Experimental Biomedicine, University of Würzburg, Würzburg, Germany

**\*For correspondence:** elmar. wolf@biozentrum.uni-wuerzburg. de

**Present address:** [†]Leibniz Institute on Aging - Fritz Lipmann Institute (FLI), Jena, Germany

**Competing interests:** The authors declare that no competing interests exist.

**Abstract** Enhanced expression of the MYC transcription factor is observed in the majority of tumors. Two seemingly conflicting models have been proposed for its function: one proposes that MYC enhances expression of all genes, while the other model suggests gene-specific regulation. Here, we have explored the hypothesis that specific gene expression profiles arise since promoters differ in affinity for MYC and high-affinity promoters are fully occupied by physiological levels of MYC. We determined cellular MYC levels and used RNA- and ChIP-sequencing to correlate promoter occupancy with gene expression at different concentrations of MYC. Mathematical modeling showed that binding affinities for interactions of MYC with DNA and with core promoter-bound factors, such as WDR5, are sufficient to explain promoter occupancies observed in vivo. Importantly, promoter affinity stratifies different biological processes that are regulated by MYC, explaining why tumor-specific MYC levels induce specific gene expression programs and alter defined biological properties of cells.

## Introduction

Deregulated expression of one of the three members of the MYC gene family (MYC, MYCN or MYCL) is observed in the majority of human tumors (*Dang, 2012*; *Meyer and Penn, 2008*). A broad body of evidence establishes that deregulated MYC expression causally contributes to multiple aspects of tumor development and that tumors depend on enhanced expression of MYC for growth and survival (*Casey et al., 2016*; *Soucek et al., 2013*).

MYC proteins are transcription factors that bind DNA as part of several protein complexes; best understood is a dimeric complex of MYC with its partner protein MAX that activates transcription via binding to a specific DNA motif termed E-box with a consensus sequence of CACGTG (*Conacci-Sorrell et al., 2014*). To repress transcription, the binary MYC/MAX complex associates with a zinc finger protein termed MIZ1 (*Wiese et al., 2013*). ChIP (chromatin-immunoprecipitation) (*Fernandez et al., 2003*) and ChIP-sequencing experiments (*Lin et al., 2012*; *Nie et al., 2012*; *Sabò et al., 2014*) demonstrate the presence of MYC on virtually all promoters with an open chromatin structure as well as on thousands of enhancers and intergenic sites in multiple cell types, raising the question what the functional relevance of this broad binding might be.

**eLife digest** Genes with the potential to cause tumors and cancer are commonly called oncogenes. One example of an oncogene encodes for a protein called MYC and many tumors contain high levels of this protein. MYC is a transcription factor and studies of aggressive tumors suggested that, like most other transcription factors, MYC binds to and regulates the activity of a small number of genes in tumors. However, other studies went on to show that MYC actually binds to thousands of genes and somehow only regulates a subset of them during tumor development.

Lorenzin et al. set out to understand how this process works by generating human cells in which the concentration of MYC protein could be altered. In the experiments, the concentration was varied from normal healthy levels to the high levels found in aggressive tumors. The amount of MYC bound to genes and the extent to which it activated the genes inside these cells was also measured.

Lorenzin et al. found that increasing MYC levels from normal to tumor-specific levels did not affect MYC binding at genes where the transcription factor was already strongly bound in normal cells. Rather, MYC binding increased only at genes that were weakly bound in normal cells. Consistent with this observation, only genes at which MYC was weakly bound in normal cells were activated by increasing MYC levels. This observation suggests that increasing the concentration of MYC protein from normal to tumor-specific levels "fills up" previously empty binding sites around these genes with the transcription factor.

Lorenzin et al. also used mathematical modeling to understand how the concentrations of MYC in normal and tumor cells might explain how MYC behaves in cells. Together, the results imply that the MYC transcription factor regulates distinct sets of genes in normal and tumor cells according to how much MYC is present. Further studies may show that the altered regulation of a tumor-specific set of genes is important for tumor development and could use this new information to identify new targets for treating MYC-driven tumors.

Given this global binding pattern, it is surprising that MYC-driven tumors can be recognized by a specific set of up- and down-regulated MYC target genes that holds considerable prognostic and therapeutic value (*Sabò et al., 2014*; *Walz et al., 2014*). One hypothesis to explain this observation suggests that MYC proteins globally enhance transcription. This has been termed the general amplifier model (*Lin et al., 2012*; *Nie et al., 2012*) and is supported by observations that MYC can cause an increase in total RNA and mRNA levels (*Grandori et al., 2005*; *Hsu et al., 2015*; *Lin et al., 2012*; *Nie et al., 2012*). In this model, specific gene expression patterns arise indirectly due to feedback and feedforward loops induced by general amplification. The alternative viewpoint suggests that MYC proteins regulate specific genes and that global changes in RNA and mRNA levels occur indirectly as a consequence of MYC-driven cell growth (*Sabò and Amati, 2014*). To explain the contrast between global binding and specific gene regulation, the latter model proposes that much of MYC binding to chromatin is non-productive in terms of transcriptional regulation (*Kress et al., 2015*).

We show here that the divergent models can be reconciled with experimental observations without the need to invoke productive and non-productive modes of DNA binding. We analyzed U2OS cells that express a doxycycline-inducible allele of MYC (*Elkon et al., 2015*; *Walz et al., 2014*) and characterized DNA binding and gene expression patterns at different levels of MYC. We showed previously that doxycycline-induced overexpression of MYC in these cells establishes a gene expression signature, which closely resembles multiple established signatures of MYC target genes and identifies expression signatures of patients with MYC amplification in tumors (*Walz et al., 2014*). Hence, these cells represent a simple model system, in which the effect of physiological and tumor-specific MYC levels can be compared and provide a tool to elucidate the mechanism(s) by which activation of a globally binding transcription factor can result in regulation of specific and functionally relevant gene expression patterns.

# Results

## MYC binding to chromatin appears saturated at certain sites

To determine the effect of changes in MYC levels on DNA binding and gene expression, we have previously engineered U2OS cells to express a doxycycline-inducible allele of MYC (*Figure 1A*). We chose U2OS cells because they have relatively low levels of endogenous MYC, despite being tumor cells. To illustrate this point, we determined MYC levels in a number of normal and transformed cells. Lysates of equal numbers of exponentially growing cells were probed by immunoblotting (*Figure 1B*). The results show that U2OS cells express levels of MYC that are comparable to non-transformed cells (IMEC, HMLE, MCF10A) and lower than those found in other tumor cell lines (HeLa and HCT116). Notably, prolonged exposure (>3 days) to doxycycline and hence long-term ectopic expression of MYC induces apoptosis in U2OS cells (*Walz et al., 2014*); therefore, all subsequent analyses were performed 28–30 hr after addition of doxycycline.

ChIP-sequencing of MYC in U2OS$^{Tet-on}$ cells had previously shown that endogenous MYC binds to about 5,500 promoters and that this number increases to about 8,400 MYC-bound promoters upon addition of doxycycline (*Walz et al., 2014*). Different peak calling programs (MACS14, SICER, CCAT) with default parameters result in very similar peak numbers (*Figure 1—figure supplement 1A,B*). Reducing the stringency of peak calling resulted in a moderate increase in peak number but a large increase in the number of negative peaks, suggesting that an analysis using default parameters does not overlook a large number of significant peaks (*Figure 1—figure supplement 1C*). In agreement with reports from other systems, we concluded that the promoters of the majority of all expressed genes are bound by MYC. Surprisingly, individual promoters showed a wide range of occupancies for endogenous MYC (*Figure 1C–E*), whereas the differences in occupancy by MYC among promoters were much smaller after induction with doxycycline (*Figure 1D,E*). Promoters, which are least strongly bound by endogenous MYC recruit most MYC upon overexpression (i.e. *VEGFA*, *Figure 1E*), whereas the most strongly bound genes recruit no additional MYC (i.e. *RPL8*). This prompted us to analyze whether the anti-correlation between occupancy by endogenous MYC and recruitment of exogenous MYC is evident globally (*Figure 1F*). To this end, we determined the relative MYC recruitment at each bound promoter by calculating the fold-change of MYC occupancy in cells with exogenous and endogenous MYC levels. Genes were ranked according to these MYC recruitment values and plotted against the respective occupancy of endogenous MYC (*Figure 1F*, blue dots). Strikingly, genes, which are most weakly bound (mean: 89 tags), recruit MYC most strongly (5.3-fold), whereas the most strongly bound genes (mean: 580 tags) show on average no further MYC recruitment. Importantly, when MYC occupancy at exogenous MYC levels is analyzed (*Figure 1F*, orange dots), all bins of genes are bound to a high extent. One way to explain this observation is the hypothesis that genes strongly bound by endogenous MYC levels are fully occupied ('saturated') and hence exogenous MYC is preferentially recruited to weakly bound genes.

To test whether this explanation is correct, we made use of the observation that MYC/MAX heterodimers compete with MXD/MAX heterodimers and MAX homodimers for binding to their target sites (*Conacci-Sorrell et al., 2014*). If promoters were fully occupied by MYC/MAX heterodimers, one would predict that MXD proteins are completely displaced from these promoters. We tested this prediction by ChIP assays on four genes using an anti-MXD6 (MNT) antibody (*Figure 1G*). Binding of MXD6 was barely detectable above background for the two genes (*NPM1, NCL*), which are strongly occupied by endogenous MYC, and did not further decrease upon addition of doxycycline. In contrast, MXD6 occupancy was higher for two genes (*HSPBAP1, FBX32*), which are poorly bound by endogenous MYC, but strongly decreased upon induction of MYC. Taken together, the easiest model to explain the data is to suggest that a few hundred of promoters are saturated by endogenous MYC in U2OS cells and that overexpression of MYC leads to saturation of the majority of MYC-binding sites in promoters (*Figure 1F*).

## Absolute quantification of nuclear MYC allows an estimate of MYC binding affinities

To understand whether the number of MYC molecules per cell in U2OS cells is able to saturate the numerous genomic binding sites, we quantified the absolute expression levels of MYC. A carboxy-terminal fragment of human MYC comprising amino acids 353 to 434 was purified to homogeneity

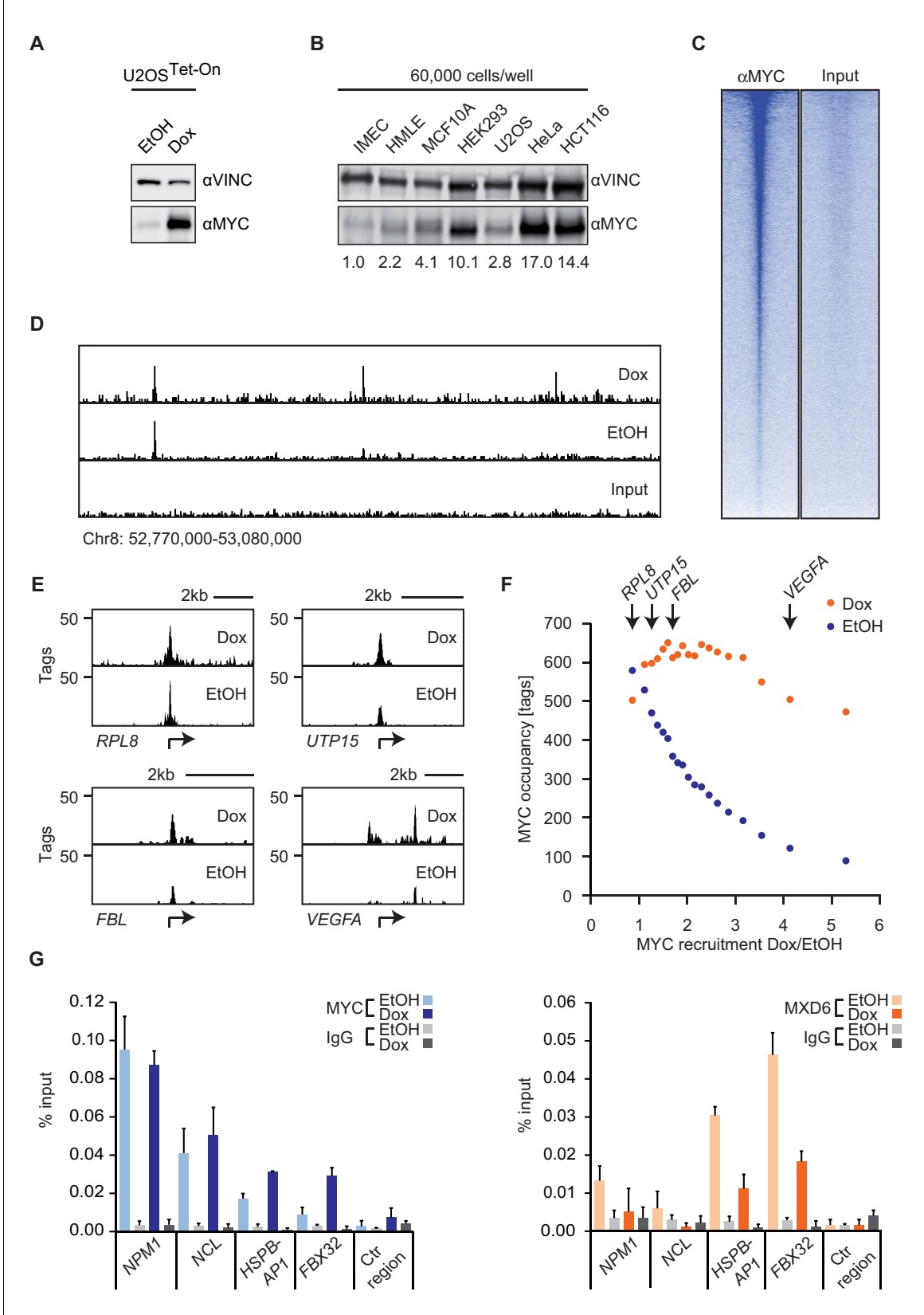

**Figure 1.** MYC saturates certain binding sites. (A) Immunoblot of MYC and Vinculin in U2OS<sup>Tet-On</sup> cells treated with EtOH or with 1 µg/ml of doxycycline. (B) Immunoblot of MYC and Vinculin in several transformed (U2OS, HeLa, HCT116) and untransformed cell lines (IMECs, HMLE, MCF10A,

*Figure 1 continued on next page*

*Figure 1 continued*

HEK293). For each sample, 60,000 cells were loaded. A quantification is shown at the bottom. (C) Heat maps for binding of endogenous MYC in U2OS$^{Tet-On}$ cells to all UCSC annotated promoters in a window of 5 kb around the transcriptional start site (TSS). Input is shown as control and intensity of color indicates binding strength. (D) ChIP-sequencing traces of MYC for one genomic region as an example. Input is shown as control. (E) ChIP-sequencing traces of MYC for four bound genes. RPL8 is a ribosomal protein, UTP15 and FBL are ribosomal biogenesis factors and VEGFA takes part in cellular signaling. A scale bar is shown at the top of each browser picture. (F) Binned plot for the comparison of MYC recruitment (change in occupancy, x-axis) and MYC occupancy (y-axis) in U2OS$^{Tet-On}$ cells expressing endogenous levels of MYC (EtOH, blue dots) or overexpressing MYC (Dox, orange dots). 8,425 genes bound by MYC upon treatment with doxycycline were sorted according to MYC recruitment and divided in 20 equally sized bins. Each dot represents the average value of the bin. The bins containing the genes shown in panel E are indicated. (G) Quantitative ChIP experiments for MYC (left panel) and MXD6 (right panel) at four MYC target genes and a control region. IgG were used as control. U2OS$^{Tet-On}$ were treated either with EtOH or with 1 μg/ml doxycycline to induce exogenous MYC expression. Data are shown as mean ± standard deviation of technical triplicates.

The following figure supplement is available for figure 1:

**Figure supplement 1.** Effect of different peak calling programs and parameters on peak numbers.

and its amount was determined using spectrometric methods (*Figure 2—figure supplement 1A*). We then used defined quantities of this fragment to calibrate a series of immunoblots. After complete transfer of both, the recombinant protein and cellular MYC (*Figure 2—figure supplement 1B*), membranes were probed with the 9E10 monoclonal antibody, which recognizes an epitope (E-QKLISEEDL) corresponding to amino acids 410 to 419 of human MYC (*Figure 2A*, *Figure 2—figure supplement 1D–F*). From triplicate experiments, we estimated that U2OS cells express approximately 100,000 molecules of endogenous MYC per cell and that this number increases to approximately $3 \times 10^6$ molecules of MYC upon induction with 1 μg/ml doxycycline (see calculations in *Supplementary file 1*). We performed immunofluorescence to estimate the cell-to-cell variation in MYC levels (*Figure 2—figure supplement 2*). Induction of MYC expression by doxycycline was also observed *in situ* by immunostaining for MYC and showed that MYC is overexpressed in all cells (*Figure 2—figure supplement 2A*). Staining and quantification with three different antibodies demonstrated that endogenous MYC levels vary less than +/− 3.7-fold in 80% of all cells (EtOH, *Figure 2—figure supplement 2B,C*) and less than +/− 2.9-fold upon overexpression (doxycycline, *Figure 2—figure supplement 2D,E*). Previous estimates found that two human tumor cell lines derived from small cell lung cancer and multiple myeloma express up to 880,000 molecules of MYC per cell (*Lin et al., 2012*), confirming that upon maximal induction with doxycycline most U2OS cells express MYC levels comparable or slightly higher to levels found in human tumor cells.

We used the estimated number of cellular MYC molecules and the nuclear volume of U2OS cells (*Koch et al., 2014*) to calculate the nuclear concentration of MYC. Knowing both the nuclear MYC concentration and the occupancy of every promoter in cells at endogenous and exogenous MYC levels (ChIP-sequencing +/− doxycycline) allowed us to estimate affinities for all MYC bound promoters. We calculated the concentration of MYC required for half-maximal occupancy of each promoter (EC$_{50}$) and used it as a measure for the apparent binding affinity (*Figure 2B*). Promoters with low EC$_{50}$ values showed a high, and those with high EC$_{50}$ values a low binding affinity toward MYC. A density plot illustrates that EC$_{50}$ values of individual promoters vary over a large concentration range (*Figure 2C*).

To better understand the functional significance of the differences in promoter affinity, we used a modified gene set enrichment (GSE) analysis (*Subramanian et al., 2005*), which uses EC$_{50}$ rather than expression values to test whether different biological processes can be stratified by promoter affinity towards MYC. In this analysis, promoters of MYC target genes that encode functionally related proteins are enriched if they exhibit similar EC$_{50}$ values. Notably, genes encoding ribosomal proteins (RPs) showed the lowest EC$_{50}$ values, followed by genes encoding proteins involved in biosynthetic processes, translation and ribosome biogenesis (*Figure 2D*; gene sets are shown in *Supplementary file 2*). These genes are thought to comprise a core signature of highly expressed MYC target genes (*Ji et al., 2011*). At the other extreme, genes encoding metabolite transporters, G-protein coupled receptors and genes involved in TGF-beta signaling and in the response to hypoxia are among those with the highest EC$_{50}$ values (*Figure 2E*). We hypothesized that differences in

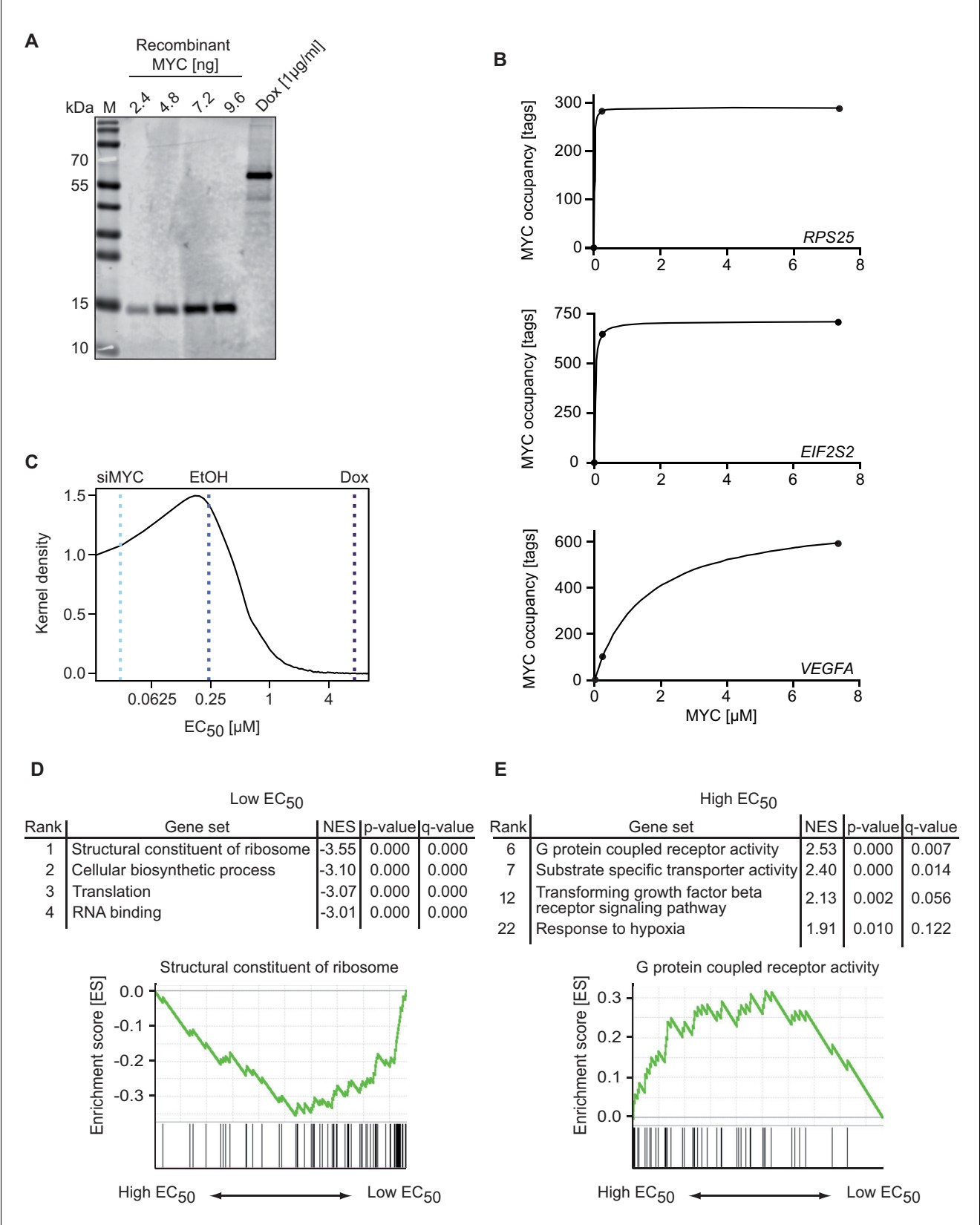

**Figure 2.** MYC binds with a wide range of affinity (EC$_{50}$ values) to target genes. (**A**) Immunoblot of MYC in U2OS$^{Tet-On}$ cells treated with 1 µg/ml doxycycline and a recombinant MYC protein fragment, which was used for absolute quantification of cellular MYC levels (M: marker). Absolute
*Figure 2 continued on next page*

*Figure 2 continued*

quantification is based on biological triplicates shown in *Figure 2—figure supplement 1D–F*. (B) Diagram of MYC occupancy calculated in ChIP-sequencing experiments of EtOH- or doxycycline-treated U2OS[Tet-On] cells (y-axis) versus the cellular MYC concentration (x-axis). The line was fitted using a Michaelis-Menten model and non-linear regression. (C) Density plot of the distribution of the $EC_{50}$ values calculated for all MYC-bound genes. Dashed lines indicate the cellular MYC concentration in uninduced (EtOH, blue line), 1 µg/ml doxycycline treated (Dox, dark blue line), or MYC-depleted (siMYC, light blue) U2OS[Tet-On] cells. (D, E) GSE analysis using the MSigDB C5 (GO gene sets) collection, of genes sorted according to $EC_{50}$ values. Enrichment plots of two gene sets enriched in the GSE analysis are shown as examples. NES: normalized enrichment score. Both, gene sets with very low (D) and very high (E) $EC_{50}$ values are shown.

The following figure supplements are available for figure 2:

**Figure supplement 1.** Quantification of MYC molecules per U2OS cell.

**Figure supplement 2.** Variation of MYC levels within the cell population demonstrates validity of the model conclusions for the majority of cells.

promoter affinity enable distinct concentrations of MYC to regulate functionally different sets of target genes.

## Binding to DNA and to WDR5 accounts for high promoter affinity

Given the high variation in $EC_{50}$ values, we wondered which factors account for promoter affinity towards MYC. In a complex with MAX, MYC directly contacts E-box sequences in DNA. We initially tested whether the known DNA binding properties of MYC/MAX heterodimers can explain the $EC_{50}$ values measured in ChIP-sequencing experiments. The heterodimer makes both base-specific contacts and contacts to the phosphate backbone of DNA and hence binds to canonical E-boxes (CACGTG), non-canonical E-boxes (CANNTG) and DNA with a random sequence (*Nair and Burley, 2003*). We modeled the binding behavior of MYC assuming canonical E-boxes and unspecific binding sites at random DNA sequences, which are in excess over the canonical E-boxes (model 1, *Figure 3A*, Appendix 1). This model ignores competition of MYC/MAX heterodimers with other E-box binding proteins with the same binding specificity, such as MXD/MAX complexes, as well as MITF, USF and TFE-3 (*Conacci-Sorrell et al., 2014*). We explored how the occupancy of canonical E-boxes in the experimentally determined range of cellular MYC concentrations depends on the dissociation constants of MYC reversibly binding to canonical E-boxes ($K_{Ebox}$) or unspecific binding sites ($K_{NNNNNN}$) (*Figure 3B*; note that all simulations show steady state solutions discussed in Appendix 1). The simulations demonstrate that occupancies of canonical E-boxes above 90% can be observed for certain combinations of dissociation constants. In contrast, occupancy of unspecific binding sites is much smaller than 1% in the considered parameter space (*Figure 3C*) illustrating that the number of unspecific binding sites strongly exceeds the number of canonical E-boxes (Appendix 1). The model assumes that the entire genome is accessible and not blocked due to heterochromatin formation. An extended analysis investigating the impact of heterochromatin (Appendix 2) confirms that our assumption on genome accessibility hardly affects our presented results and conclusions. Values of dissociation constants have been previously determined for canonical E-boxes and for DNA with a non-E-box sequence. We fixed the dissociation constants in our subsequent model analyses to one pair of published values (*Guo et al., 2014*) (see red and blue lines in *Figure 3B,C*). The respective occupancies of canonical E-boxes and unspecific binding sites in the experimentally determined range of total MYC abundance are shown in *Figure 3D* (red and blue lines, respectively). Under these assumptions, MYC/MAX complexes are predicted to bind canonical E-boxes with a calculated $EC_{50}$ value of $1\times10^2$ µM (*Figure 3D*). Comparison with the experimentally determined $EC_{50}$ values showed that only 363 of 8,425 promoters show this or a higher $EC_{50}$ value.

The majority of E-boxes located in promoters showed considerably lower $EC_{50}$ values than those predicted by the affinity to E-box-DNA, indicating that binding to DNA alone is not sufficient to account for the chromatin occupancy of MYC observed by ChIP-sequencing. Model simulations predict that a reduction of the dissociation constant of MYC and E-boxes by about one order of magnitude shifts the $EC_{50}$ value into the measured range of MYC molecules (*Figure 3E*). To explore potential underlying molecular mechanisms, we searched for features, which identify promoters with

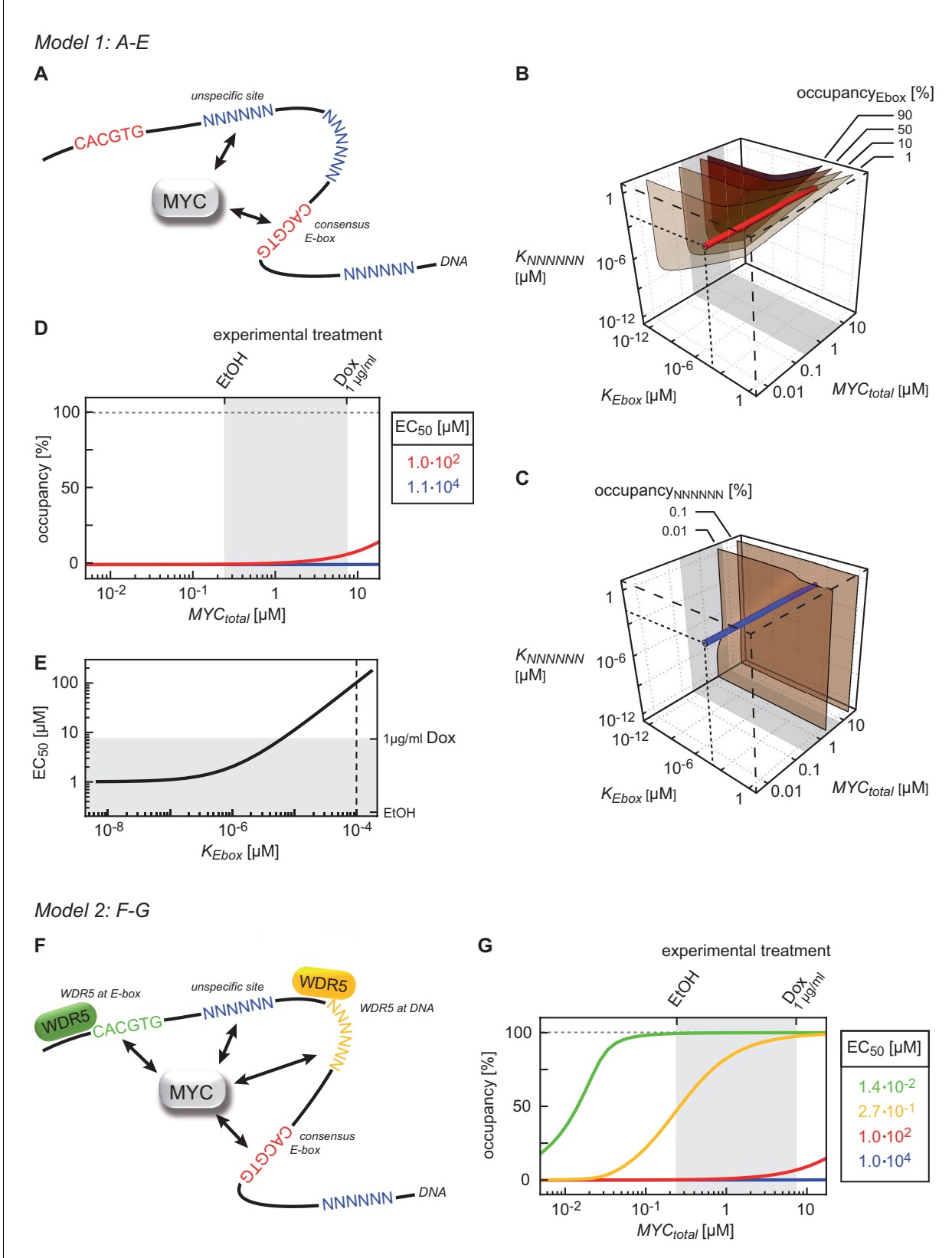

**Figure 3.** Binding behavior of MYC in U2OS cells analyzed by mathematical modeling. (**A**) Schematic representation of model 1. For details see Appendix 1. (**B**, **C**) Plot illustrating regions of occupancy in the parameter space of dissociation constants $K_{Ebox}$ and $K_{NNNNNN}$ as well as total amount of

*Figure 3 continued*

MYC. The grey area indicates the experimentally available concentration range, that is, EtOH to 1 µg/ml doxycycline (Dox) treatment. Regions of 1, 10, 50 and 90% occupancy of E-boxes (B) as well as 0.01% and 0.1% occupancy of NNNNNN sequences (C) are shown. The red line (B) and blue line (C) give the combination of the particular dissociation constants published by *Guo et al. (2014)*. (D) Simulations show that occupancy of E-boxes (red line) by MYC is less than 10% in the measured range of MYC (grey area) while occupancy of NNNNNN (blue line) is below 1%. The EC$_{50}$, which is the concentration of total MYC to obtain 50% occupancy, is calculated to be 1x10$^2$ µM for E-boxes. (E) The EC$_{50}$ of E-boxes (1x10$^2$ µM estimated in D) can be reduced by decreasing the value of $K_{Ebox}$. Simulations predict that a reduction of $K_{Ebox}$ (value published by *Guo et al., 2014*, is indicated by the dashed line) by about one order of magnitude already shifts the EC$_{50}$ into the measured range of MYC (grey area). (F) Schematic representation of model 2. For details see Appendix 1. (G) In model 2, a reduction of the apparent dissociation constant of MYC and E-boxes as well as that of MYC and unspecific DNA sites are assumed by means of additional regulatory proteins such as WDR5. In the presence of WDR5, occupancy of E-boxes by MYC is above 95% (green line, EC$_{50}$= 1.4x10$^{-2}$ µM) and occupancy of unspecific DNA sites by MYC is above 50% (yellow line, EC$_{50}$ =2.7x10$^{-1}$ µM). Occupancy of E-boxes or unspecific DNA sites that are not bound by WDR5 (red and blue line, respectively) remain however below 10% and 1%, respectively, in the measured range of MYC (grey area).

high affinity for MYC. Consistent with the DNA binding properties, we found a strong correlation between the occupancy of endogenous MYC and the occurrence of canonical E-box sequences in the binding region (*Figure 4A*, *Figure 4—figure supplement 1A–F*). In contrast, non-consensus E-box sequences are only moderately enriched in MYC peaks and their frequency does not positively correlate with MYC binding (*Figure 4—figure supplement 1B,D-F*). In addition to canonical E-box sequences in the binding region, occupancy by endogenous MYC positively correlated with overall expression of the respective gene (*Figure 4B*) and with features of open chromatin, such as trimethylation of histone H3K4 (*Figure 4C*). This observation is in agreement with several previous reports (*Guccione et al., 2006*; *Guo et al., 2014*; *Lin et al., 2012*; *Nie et al., 2012*). Recent work has identified WDR5, a WD40-repeat-containing protein, which is a part of the MLL/SET methyltransferases that methylate H3K4 and the MOF/NSL histone acetyltransferases that acetylate histone H4, as a direct interaction partner of MYC (*Thomas et al., 2015*). MYC binds to WDR5 with a K$_D$ of 9.3 µM via MYC BoxIII (*Thomas et al., 2015*), a domain that is not part of the DNA-binding domain, suggesting that binding of MYC to WDR5 occurs independently of binding to DNA. A modified model (model 2; *Figure 3F*; see also Appendix 1) that assumes (i) that WDR5 is constantly bound to its target sites (ii) that MYC and WDR5 are free to bind to each other when both are bound to chromatin in close proximity predicts an EC$_{50}$ value of 0.014 µM for MYC occupancy of an E-box in the presence of WDR5 (*Figure 3G*). This value is lower than the one estimated experimentally for the large majority of promoters, arguing that the MYC/WDR5 interaction is of sufficient high affinity to explain the high occupancy of promoters with low EC$_{50}$ values (7,963/8,425).

The model predicts that occupancy by MYC is strongest at promoters containing both E-boxes and WDR5-binding sites but is also high for promoters bound by WDR5 but lacking E-boxes (EC$_{50}$ value of 0.27 µM; *Figure 3G*, yellow curve). This prediction could be confirmed by stratifying all MYC bound promoters in U2OS cells by these features and analyzing the individual groups for occupancy by endogenous MYC, functional annotation and expression (*Figure 4—figure supplement 2A–C*). Promoters bound by WDR5 and containing consensus E-box sequences are most strongly bound by MYC (*Figure 4—figure supplement 2A*), enriched in genes encoding for nuclear proteins (*Figure 4—figure supplement 2B*) and are associated with high expression (*Figure 4—figure supplement 2C*). The two central predictions of the model are that (i) binding sites are little occupied at cellular MYC concentrations if only the affinity of MYC to E-boxes is considered (*Figure 3G*, red curve), and that (ii) the occupancy strongly increases if the interaction of MYC with WDR5 is considered in addition (*Figure 3G*, green curve). These predictions are valid for at least 80% of cells when the cell-to-cell variation in the population is taken into account (*Figure 2—figure supplement 2F*).

To test the notion that the interaction between MYC and WDR5 is critical for the high occupancy of some promoters, we re-analyzed published ChIP-sequencing data, performed in HEK293 cells, for WDR5, wild-type MYC and a mutant allele of MYC ('MYC$^{WBM}$'), which is strongly compromised in binding to WDR5 (*Thomas et al., 2015*). In agreement with the published data, our re-analysis showed that (i) the majority of promoters is bound by WDR5, (ii) almost all binding sites of MYC in promoters overlap with WDR5 binding and (iii) disruption of the interaction between MYC and

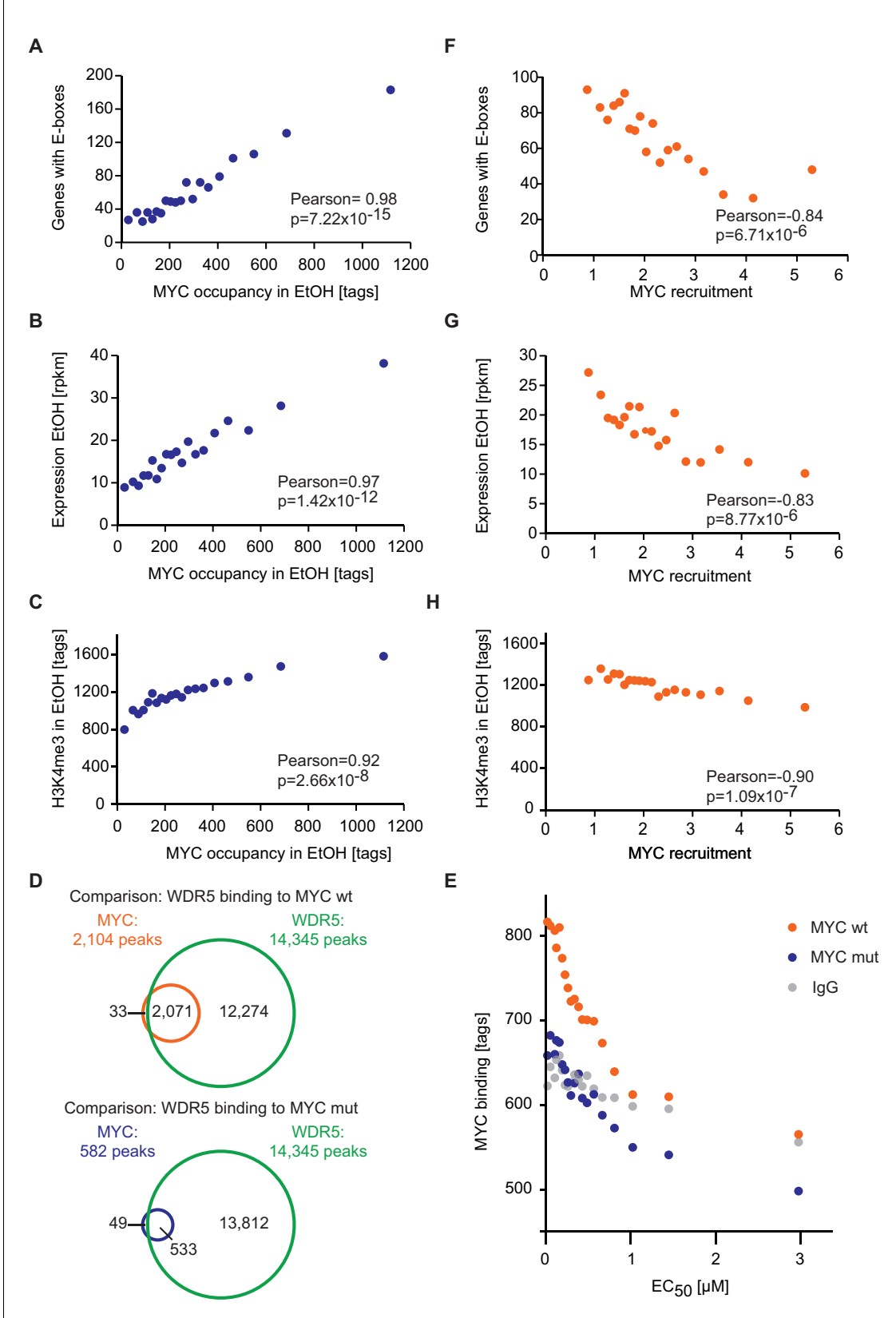

**Figure 4.** E-box occurrence, expression level and chromatin status of target genes influence MYC binding. (**A**) Binned plot for the number of genes in each bin with a canonical E-box (CACGTG) in the MYC peak versus MYC occupancy in U2OS[Tet-On] treated with EtOH. Genes were sorted according to

*Figure 4 continued on next page*

*Figure 4 continued*

MYC occupancy and divided into 20 bins. Each dot represents the average of 422 genes. (B) Binned plot as in A, but with the mRNA expression of the respective gene. Reads per kilobase per million (rpkm) are shown on the y-axis. (C) Binned plot as in A, but with H3K4me3 status of the respective gene. (D) Venn diagram displaying the promoter-close (+/− 5 kb) binding site overlap of WDR5, wild-type MYC (top), and a MYC mutant, which is compromised in binding to WDR5 ('MYC mut', bottom). Both, wild-type and mutant MYC were fused to a Flag epitope and stably expressed in HEK293 cells by *Thomas et al. (2015)*. (E) Binned plot for MYC binding vs $EC_{50}$ values. Genes bound by MYC in U2OS cells were sorted according to their $EC_{50}$ values and correlated to average occupancy of a MYC mutant compromised in binding to WDR5 (blue dots) or wildtype MYC (orange dots). Values for IgG are shown as a background control (grey dots). Panel D and E are re-analyses based on published data (*Thomas et al., 2015*). (F) Binned plot for the number of genes in each bin having a canonical E-box (CACGTG) in the MYC peak versus MYC recruitment. Genes were sorted according to MYC recruitment and divided in 20 bins. Each bin represents 422 genes. (G) Binned plot as in F, but the mRNA expression of the respective gene. Reads per kilobase per million (rpkm) is shown on the y-axis. (H) Binned plot as in F, but H3K4me3 status of the respective gene was analyzed.

The following figure supplements are available for figure 4:

**Figure supplement 1.** E-box occurrence and level of expression correlate with MYC binding and $EC_{50}$ values.

**Figure supplement 2.** Promoters that are bound by WDR5 and contain E-boxes are most strongly occupied by MYC.

WDR5 strongly decreases MYC binding to chromatin (*Figure 4D*). Strikingly, analysis of MYC$^{WBM}$ binding in HEK293 cells showed that eliminating MYC's ability to interact with WDR5 most strongly affects high-affinity binding sites with low $EC_{50}$ values (*Figure 4E*). We conclude that known interactions of MYC with DNA together with promoter-bound factors can account for the range of promoter occupancies observed *in vivo*.

Since target gene expression, H3K4me3 and the occurrence of E-box sequences stratify high-affinity binding sites (*Figure 4A–C*, *Figure 4—figure supplement 1G,H*), we wondered whether this correlation also holds true for MYC that is recruited to promoters upon doxycycline induction. Importantly, we found a strong anti-correlation between the recruitment of exogenous MYC on one hand and the occurrence of E-boxes (*Figure 4F*), the expression level of target genes (*Figure 4G*) and trimethylation of histone H3K4 (*Figure 4H*) on the other hand. This highlights that the major change in occupancy is due to MYC binding to low-affinity binding sites and to weakly expressed genes when MYC levels increase.

## Binding affinity determines MYC-mediated transcriptional responses at different MYC concentrations

Previously, we found that genes encoding ribosomal proteins and proteins involved in ribosomal biogenesis are not regulated in response to doxycycline-mediated MYC overexpression (*Walz et al., 2014*), supporting the hypothesis that MYC binding sites in their promoters are fully occupied by MYC at endogenous levels. To exclude the possibility that these genes are not regulated by MYC in U2OS cells, we depleted endogenous MYC in U2OS cells by siRNA and analyzed gene expression by RNA-sequencing (*Figure 5A–D*) and MYC binding by ChIP-sequencing (*Figure 5—figure supplement 1A–G*). RNA-sequencing identified 753 genes being down- and 1224 genes being significantly up-regulated (log$_2$FC < −1 or > 1, q-value < 0.01, *Figure 5B*). Expression data were analyzed by GSEA to identify functional groups of MYC-regulated genes. Strikingly, ribosomal protein genes and genes encoding proteins involved in ribosomal biogenesis are most strongly down-regulated upon siRNA treatment (*Figure 5C,D*). Accordingly, ChIP-sequencing upon depletion of MYC (*Figure 5—figure supplement 1A,C*) demonstrated global loss of MYC binding to high-affinity binding sites (*Figure 5—figure supplement 1B,D*) including promoters of ribosomal protein genes (*Figure 5—figure supplement 1E,F*). Strikingly, genes, which show the highest affinity for MYC, most strongly respond to depletion of endogenous MYC (*Figure 5—figure supplement 1G*). We confirmed our previous observations that these gene sets are not regulated when MYC is overexpressed (*Figure 5E*). To test whether similar dose-dependent transcriptional responses to MYC are also observed during tumorigenesis, we utilized data from a murine model for MYC-driven B cell lymphomagenesis (*Figure 5F–H*) (*Sabò et al., 2014*). Expression profiles are published for three different conditions, (i) resting B cells expressing only low levels of endogenous MYC ('control'), (ii) proliferating B cells expressing intermediate levels of MYC from the Eμ-MYC transgene ('pre-tumor') and (iii)

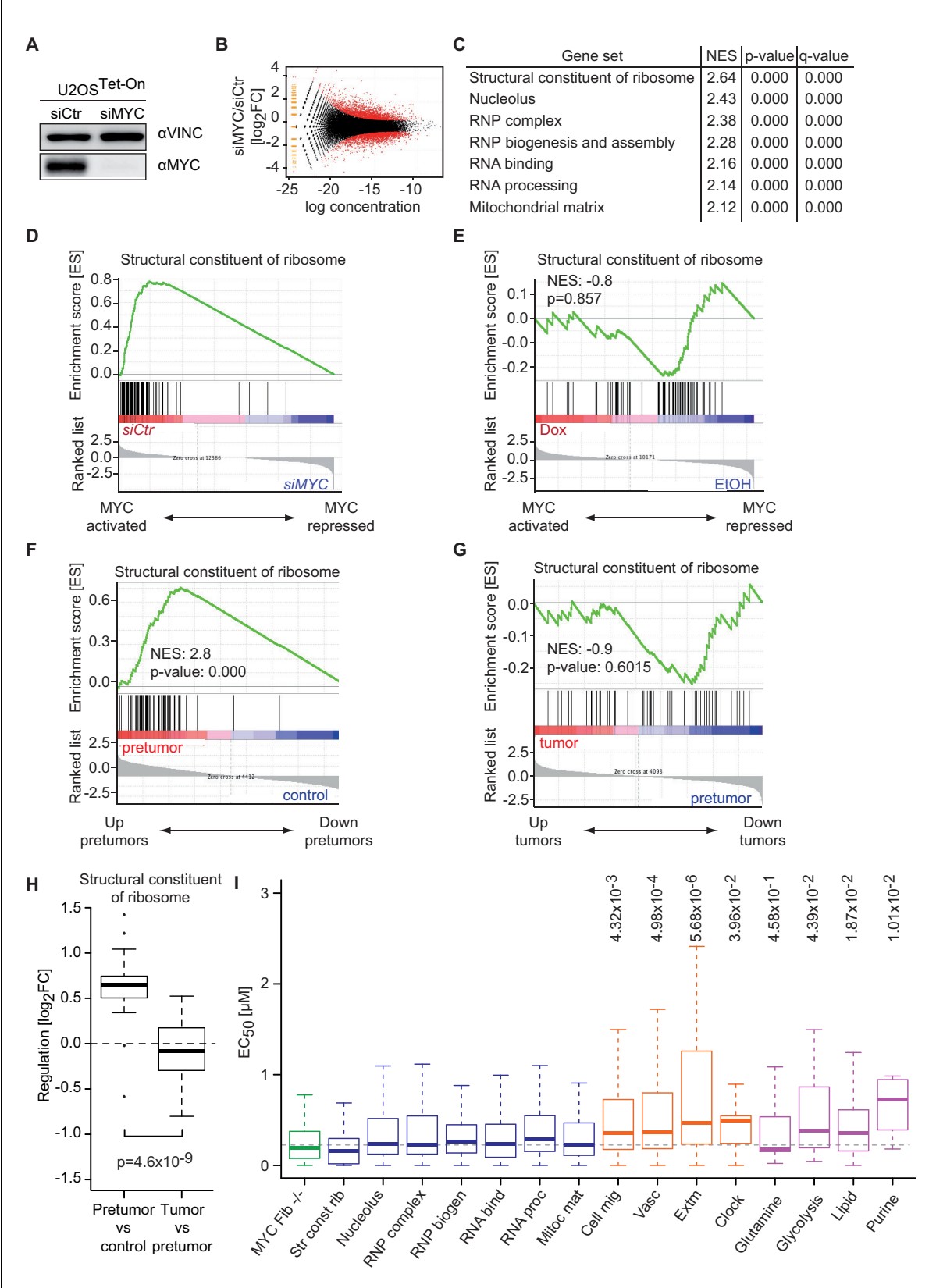

**Figure 5.** Promoter affinity for MYC stratifies functionally different gene sets. (**A**) Immunoblot of MYC upon transfection of U2OS[Tet-On] cells with the indicated siRNA. Vinculin was used as loading control. (**B**) Plot for regulation of genes upon siRNA-mediated depletion of MYC vs relative expression

*Figure 5 continued on next page*

*Figure 5 continued*

levels (known as M/A-plot). Red dots represent significantly regulated genes (q-value <0.01, q-values were estimated by Benjamini-Hoechberg-corrected p-values calculated by EdgeR). The experiment was performed in biological triplicates. (C) GSE analysis comparing gene expression profiles of siCtr- or siMYC-transfected U2OS$^{Tet-On}$ cells. NES: normalized enrichment score. (D) Enrichment plot of the 'Structural constituent of ribosome' gene set found in C upon depletion of endogenous MYC by siRNA. (E) Enrichment plot of the same gene set as in D, but gene expression profiles of U2OS$^{Tet-On}$ cells uninduced (EtOH) or induced with 1 µg/ml doxycycline were used. (F-H) Gene expression analysis comparing the gene expression profile of a ribosomal gene set in the Eµ-MYC model of B cell lymphomagenesis in three different conditions. Differential expression is shown by GSE analysis comparing a pre-tumor condition to control condition (F) and a tumor condition to a pre-tumor condition (G) or by boxplots (H). Panel F to H are re-analyses of published B cell expression profiles (*Sabò et al., 2014*). (I) Boxplots of gene sets found in panel C (blue), genes regulated upon MYC knockout in fibroblasts (green) (*Perna et al., 2012*), gene sets regulated by MYC overexpression (orange) (*Walz et al., 2014*) and selected metabolic processes (violet). The y-axis shows EC$_{50}$ values calculated as shown in *Figure 2*. p-values: One-sided Mann-Whitney-Wilcoxon test comparing the indicated GO terms to the median of 'RNP complex' (dashed line). Outliers not shown.
The following figure supplement is available for figure 5:

**Figure supplement 1.** Changes in chromatin binding upon MYC depletion correlate with transcriptional responses.

transformed B cells selected for very high MYC expression (*Sabò et al., 2014*). Strikingly, ribosomal protein genes, hence the genes with lowest EC$_{50}$ values, are strongly induced when the pre-tumor condition is compared to control cells (*Figure 5F,H*). In contrast, expression of ribosomal protein genes does not increase further when tumors are compared with pre-tumor cells (*Figure 5G,H*), although MYC protein levels increase about eight fold (*Sabò et al., 2014*). Conversely, some genes that are regulated by exogenous MYC in U2OS cells and have low-affinity promoters, like those encoding transmembrane transporters, are also regulated when comparing the tumor to pre-tumor condition in B cell lymphomagenesis (*Figure 5—figure supplement 1H,I*). Therefore, we propose that selective transcriptional responses by MYC at physiological and oncogenic levels are similar *in vivo* and *in vitro*.

To further explore the correlation between binding affinity and selective transcriptional responses, we compared median EC$_{50}$ DNA binding values of gene sets that respond to depletion of endogenous MYC to EC$_{50}$ values of gene sets that are regulated in response to MYC overexpression (*Figure 5I*). Gene sets, which are strongly regulated upon overexpression, for example, genes involved in angiogenesis and cell migration (*Figure 5I*, orange boxplots), show a significantly higher median EC$_{50}$ value than gene sets derived from the depletion of endogenous MYC (*Figure 5I*, blue boxplots). We noted that also genes involved in circadian rhythm are regulated upon MYC overexpression and showed high EC$_{50}$ values, consistent with the recent demonstration that disruption of the circadian clock is a critical transforming function of MYC (*Altman et al., 2015*). We concluded that binding affinity determines specific MYC-mediated transcriptional responses at physiological and oncogenic MYC concentrations. To further support this conclusion, we analyzed expression data from embryonic fibroblasts isolated from mice that carry floxed *MYC* alleles (*Perna et al., 2012*). Strikingly, EC$_{50}$ values of genes that are down-regulated upon deletion of endogenous MYC are very low (*Figure 5I*, green boxplot) confirming that, as in U2OS cells, at physiological levels MYC regulates genes with the highest binding affinity. In view of reports, which demonstrate that altering cellular metabolism is a transforming mechanism of MYC (*Stine et al., 2015*), we analyzed the affinity of promoters of genes encoding proteins involved in selected metabolic processes and indeed observed elevated EC$_{50}$ values for lipid- and purine metabolism and potentially glycolysis (*Figure 5I*, violet boxplots).

## Increasing amounts of MYC gradually shift transcriptional programs

Our data suggest that MYC regulates different sets of genes at physiological and oncogenic levels and that these gene sets can be stratified by promoter affinity. We wondered, if there are indeed two distinct modes of regulation or if MYC-mediated responses rather vary continually upon changes in cellular MYC concentrations. To test this, we gradually manipulated MYC levels in U2OS$^{Tet-On}$ cells by titrating the amount of doxycycline and calculated the absolute number of MYC molecules in triplicate experiments (*Figure 6A*, *Figure 2—figure supplement 1D–F*). Analysis of gene expression profiles at five different concentrations of doxycycline revealed different response patterns for

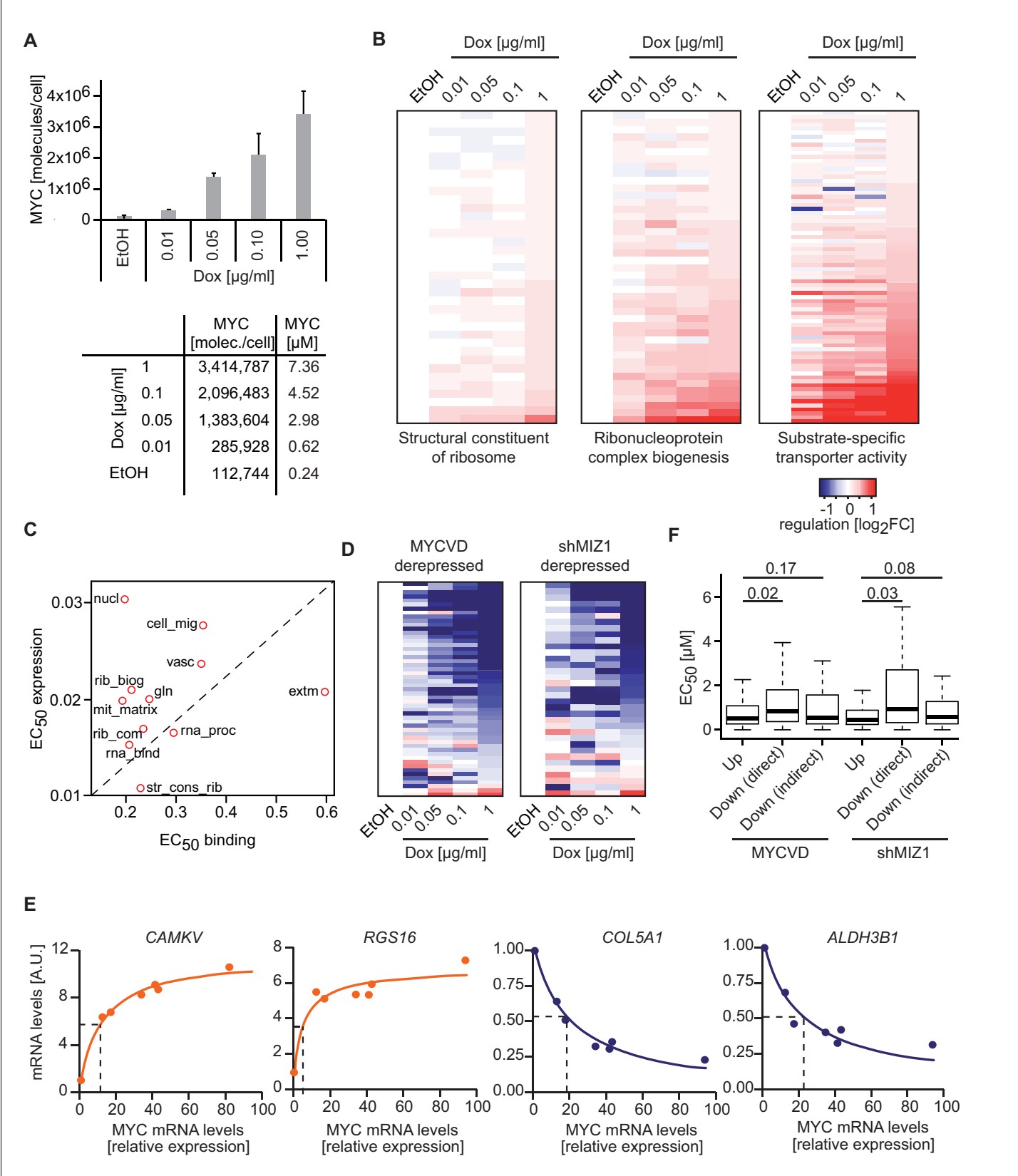

**Figure 6.** MYC levels determine the regulation of different transcriptional response programs. (**A**) Bar plot indicating MYC levels in U2OS^Tet-On cells treated with EtOH or with different concentrations of doxycycline. Data are shown as mean ± standard deviation of three independent biological

*Figure 6 continued on next page*

*Figure 6 continued*

replicates, which are shown in *Figure 2—figure supplement 1D–F*. Table of calculated MYC molecules and concentration per cell (see *Supplementary file 1* for calculation). (B) Heat maps for regulation of genes belonging to the three gene sets (Structural constituent of ribosome: GO_0003735; Ribonucleoprotein complex biogenesis: GO_0022613; substrate-specific transporter activity: GO_0022892) identified by GSEA in *Figure 2D,E* and *Figure 5C*. For all conditions, $\log_2$FC was calculated relative to the EtOH sample and only transactivated genes are shown. *Supplementary file 2* contains gene set lists used in this study. (C) Plot for median $EC_{50}$ values for binding and regulation of gene sets from *Figure 5I*. $EC_{50}$ values for regulation were calculated using RNA-sequencing data from the MYC depletion (*Figure 5A*) and the MYC titration experiments (*Figure 6A*). (D) Heat maps documenting the regulation of genes down-regulated by wild-type MYC and derepressed by $MYC^{V394D}$ or in cells depleted of MIZ1. $\log_2$FC calculated as in B. (E) qRT-PCR analysis from $U2OS^{Tet-On}$ induced for MYC overexpression with increasing doxycycline concentrations. Both, MYC mRNA levels (x-axis) and mRNA levels of MYC target genes (y-axis) are plotted. Dashed lines indicate half-maximal gene regulation. Please note the differences in MYC expression needed for half maximal target gene regulation between transactivated (orange curve) and repressed (blue curve) genes. The x-axis shows MYC mRNA levels relative to the EtOH control. (F) Boxplots of genes repressed by wild-type MYC but not by $MYC^{V394D}$ (MYCVD, 'Down (direct)') or by wild-type MYC when MIZ1 is depleted via shRNA (shMIZ1, 'Down (direct)'). MYC-activated genes ('Up') and genes, which are repressed in the absence of a functional MYC/MIZ1 complex ('Down (indirect)') were used as reference gene sets. p-values (p), One-sided Mann-Whitney-Wilcoxon test. Outliers are not shown in the plot.

different gene sets (*Figure 6B*), ranging from hardly any response to MYC overexpression ('structural constituent of ribosome'), medium-level responses ('ribonucleoprotein complex biogenesis'), to gene sets, which responded predominantly at very high MYC levels ('substrate-specific transporter activity'). The RNA-sequencing data enabled us to calculate $EC_{50}$ values for regulation and to correlate them with $EC_{50}$ values for binding. Importantly, the observed differences in dose response correlate with the promoter affinity of the genes involved in these processes (*Figure 6C*). We conclude that differences in promoter affinity can account for the distinct transcriptional response programs observed in biological systems exhibiting different MYC expression levels (*Figure 7*).

## Repressed genes exhibit low affinity for MYC

Finally, a prominent discrepancy among reports analyzing the transcriptional consequences of MYC level manipulation is the presence or absence of direct MYC-mediated repression (*Lin et al., 2012*; *Nie et al., 2012*; *Sabò et al., 2014*; *Walz et al., 2014*). As direct repression is most apparent at high levels of MYC (*Wiese et al., 2015*), we hypothesized that differences in MYC expression levels might explain this seeming contradiction. Direct MYC-mediated repression largely depends on complex formation with MIZ1 (*Peukert et al., 1997*), which is strongly reduced when expressing a point mutant of MYC ($MYC^{V394D}$). By using previously published datasets (*Walz et al., 2014*), we identified direct MYC-repressed sets of genes assuming that they are re-expressed if MIZ1 is depleted (shMIZ1) or if $MYC^{V394D}$ instead of wild type MYC is expressed. Interestingly MYC/MIZ1-mediated repression takes place at high levels of MYC, both in a global analysis (*Figure 6D*) as well as for individual genes (*Figure 6E*). Accordingly, genes directly repressed by MYC ('Down (direct)' in *Figure 6F*), show significantly higher $EC_{50}$ values than transactivated genes ('Up'), whereas indirectly repressed genes ('Down (indirect)' in *Figure 6F*) show affinities comparable to that of transactivated genes. We concluded that direct MYC-mediated repression is a predominant form of transcriptional regulation at high (oncogenic) levels of MYC, which may not be reached during normal cell growth (*Figure 7A*).

## Discussion

The aim of this study was to understand whether an analysis of MYC function at different protein levels in combination with mathematical modeling can reconcile two seemingly conflicting observations: Whereas MYC regulates specific genes in some biological settings ('specifier model'), in others MYC enhances expression of all genes transcribed by RNA polymerase II ('amplifier model'). Our study has led to two major conclusions.

The first conclusion is that direct interaction with DNA alone is not sufficient to account for the observed occupancies of core promoters by MYC/MAX complexes. This result is in agreement with experimental observations showing that the positions occupied by MYC/MAX complexes across the human genome closely correlate with the RNA polymerase II transcription machinery even on promoters, which do not contain E-boxes (*Guccione et al., 2006*; *Guo et al., 2014*). The

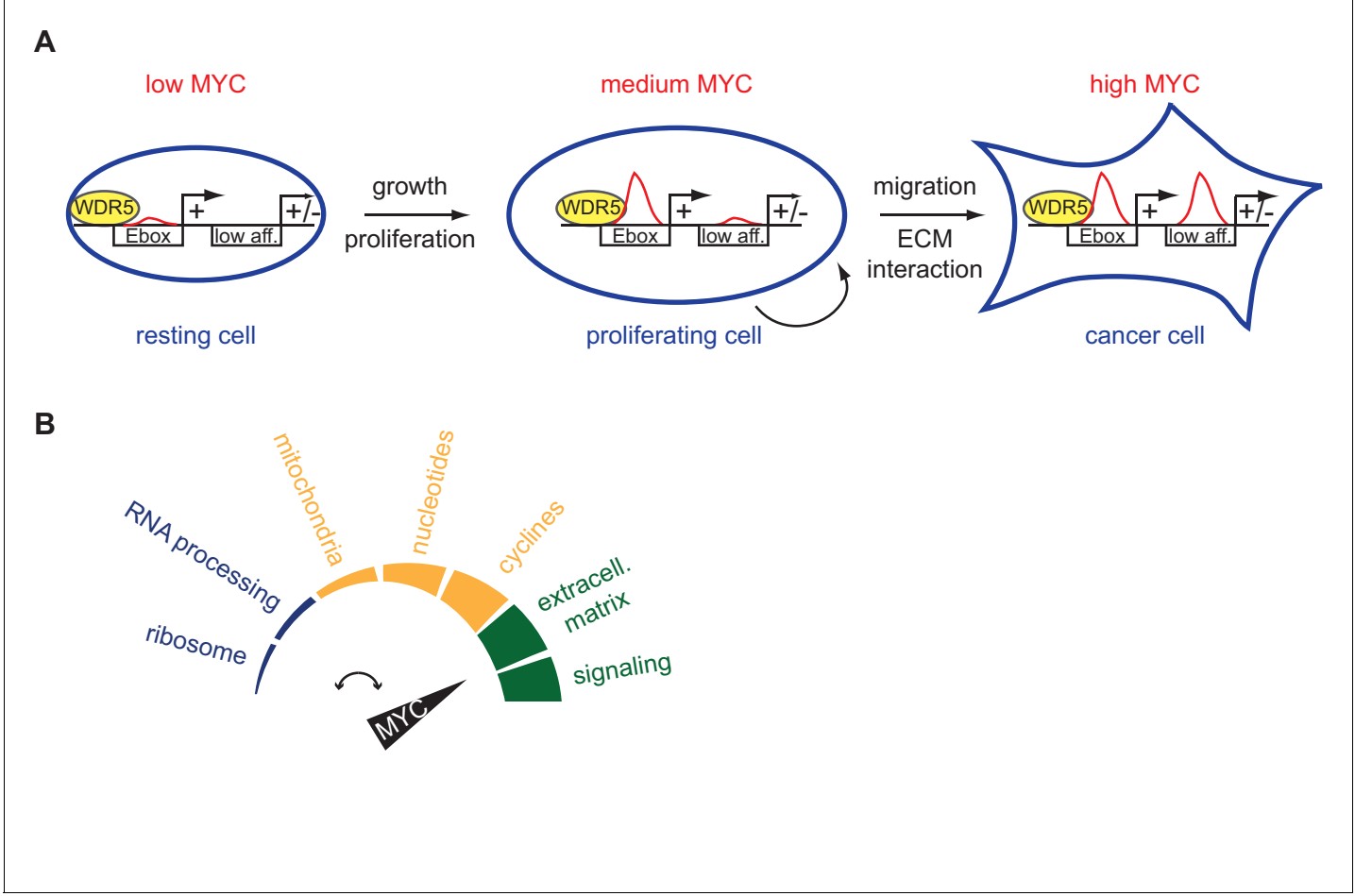

**Figure 7.** Schematic models. (**A**) High-affinity binding sites, for instance characterized by E-boxes and WDR5 binding, are already highly occupied at physiological MYC levels (medium MYC) in dividing cells. At oncogenic concentration (high MYC) also low-affinity (low aff.) binding sites can become occupied by MYC and regulation of these low-affinity genes is indicative for transformation (ECM: extracellular matrix). (**B**) MYC regulates distinct biological processes at different cellular concentrations.

conclusion is based on a modeling approach that assumes two classes of binding sites, a high-affinity class of binding sites with consensus E-boxes and a second class of binding sites containing non-specific DNA. This assumption is an over-simplification, since MYC/MAX complexes have low affinity for binding sites containing non-consensus E-box sequences, which exceeds the affinity for non-specific DNA (*Blackwell et al., 1993*) (*Figure 4—figure supplement 1E,F*). Furthermore, multiple other transcription factors recognize the same consensus sequence as MYC/MAX proteins, suggesting that there is significant competition for E-box binding among these complexes in cells (*Wolf et al., 2015*). As a consequence, the current model overestimates the fraction of MYC/MAX complexes bound to consensus E-boxes. These considerations reinforce the conclusion that DNA binding alone cannot account for the high occupancies of core promoters by MYC/MAX complexes in cells.

The mathematical analysis shows that the interaction of MYC/MAX with promoter-bound WDR5 can strongly enhance occupancy of core promoters. Consistent with this finding, a mutant allele of MYC unable to bind to WDR5 shows much reduced binding to high-affinity promoters. MYC does not only interact with WDR5, but with several components of the transcription machinery including P-TEFb, TBP and TRRAP (*Bouchard et al., 2004*; *Eberhardy and Farnham, 2002*; *Hateboer et al., 1993*; *McMahon et al., 1998*; *Rahl et al., 2010*). Additionally, we recently identified the interaction of another conserved domain of MYC, MYC BoxI, with the PAF1C complex that is located at core

promoters (*Jaenicke et al., 2016*). We restricted our analysis to WDR5, since no dissociation constants for other protein/protein interactions of MYC have been published. However, MYC high-affinity binding sites can be observed at genomic regions, which are not in proximity to WDR5 binding sites, suggesting that other factors are involved in establishing MYC genomic binding pattern. Future ChIP-sequencing studies of MYC upon depletion of MYC interaction proteins will clarify the impact of individual factors on chromatin binding of MYC.

Most likely, therefore, genome-wide occupancies of MYC/MAX complexes result both from interactions of MYC/MAX complexes with DNA and from protein/protein interactions with resident chromatin proteins. We propose that the dependence of MYC/MAX chromatin binding on interactions with proteins that are themselves influenced by existing gene expression patterns at different physiological states of cells, contributes to specific gene regulation by MYC even though MYC′s mechanism of action is the same at all promoters (as an example, see *Liu et al. (2015)*). Our analysis leads to the hypothesis that conserved protein/protein interactions of MYC are critical for targeting MYC/MAX complexes to specific classes of binding sites such as core promoters *in vivo*.

The second conclusion that can be drawn from our data is that promoter affinity for MYC confers specificity to gene regulation. Our results show that promoters differ in their apparent affinity for MYC and demonstrate a close correlation between apparent promoter affinity measured by ChIP-sequencing and response to different concentrations of MYC measured by RNA-sequencing at different MYC levels. This correlation leads to the conclusion that a gene is regulated by MYC in a given experimental setting, if the occupancy of its promoter (or a relevant enhancer) by MYC changes in this setting. Importantly, measurements of MYC levels and modeling approaches show that promoters can be saturated by levels of MYC that are reached in proliferating cells (*Figure 7A*). As a result, expression of genes may not respond to MYC in a certain experimental setting (such as a doxycycline-mediated induction of high MYC levels) since the promoters are fully occupied before addition of doxycycline. This is likely to be true for strongly transcribed genes as many ribosomal protein genes and genes encoding proteins involved in ribosome biogenesis and translation. The experimentally observed specificity in gene regulation can therefore be explained without the need to invoke productive and non-productive modes of interaction of MYC with chromatin.

Consistent with this notion, the analysis of MYC-driven lymphomagenesis shows that MYC can contribute to oncogenic transformation via two different mechanisms: low levels of constitutive MYC enhance expression of genes with high-affinity promoters, such as genes encoding proteins involved in ribosome function. This is seen when pre-tumorigenic B-cells are compared with control lymphocytes. MYC levels further increase at the transition to frank lymphomas and there is clear evidence for further increases in MYC levels during progression of several tumor entities, as for example in APC-mutant colorectal tumors (*Kress et al., 2011*; *Myant and Sansom, 2011*; *Siemens et al., 2011*). This additional increase in MYC levels will enhance occupancy of promoters that are lowly occupied in normal and proliferating cells. Since differences in apparent affinity and in $EC_{50}$ values stratify classes of promoters of genes that encode proteins that differ in function, we propose that this will alter the physiological and metabolic state (*Stine et al., 2015*) of the tumor cells (*Figure 7B*). Notably, promoters with the lowest apparent affinity are found in genes that encode proteins involved in nutrient transport, G-protein coupled receptors and in the response to hypoxia. Expression of these genes responds to further increases in MYC levels arguing that they are the basis for the selective pressure for increases in MYC levels. Since many tumors depend on ('are addicted to') high MYC levels, inhibition of proteins encoded by this class of target genes can be a rational strategy for selective inhibition of MYC-dependent tumor growth.

Finally, we note that virtually all genes that are repressed by MYC/MAX in complex with MIZ1 display low apparent affinities and high $EC_{50}$ values, suggesting that MIZ1-mediated repression occurs predominantly at high MYC levels. MIZ1-mediated repression is required for tumor formation in MYC-driven lymphoma (*van Riggelen et al., 2010*) and medulloblastoma (*Vo et al., 2016*), suggesting that targeting the complex may provide a second strategy with a significant therapeutic window for MYC-dependent tumors.

## Materials and methods

### Cell culture and transfection

U2OS, HeLa, HEK293 and HCT116 cells were cultivated in DMEM (Sigma-Aldrich, St. Louis, USA) supplemented with 10% FCS (EMD Millipore, Billerica, USA) and 1% penicillin/streptomycin (Sigma-Aldrich). IMECs and HMLE cells were cultivated in DMEM/F12 (co) supplemented with 1% penicillin/streptomycin (Sigma-Aldrich), 10 µg/ml insulin (Sigma-Aldrich), 0.5 µg/ml hydrocortisone (Sigma-Aldrich) and 20 ng/ml EGF (Life Technologies, Carlsbad, CA). MCF10A cells were cultivated in DMEM/F12 (Thermo Fisher Scientific, Waltham, USA) supplemented with 5% horse serum (Sigma-Aldrich), 10 µg/ml insulin (Sigma-Aldrich), 0.5 µg/ml hydrocortisone (Sigma-Aldrich) and 20 ng/ml EGF (Life Technologies) and 100 ng/ml choleratoxin (Sigma-Aldrich). Cell lines were tested for mycoplasma and authenticated by STR profiling (HCT116, MCF10A, HeLa, U2OS). They were not found to be on the list of commonly misidentified cell lines (International Cell Line Authentication Committee, with the exception of HEK cells) and received from research labs, companies (HCT116: LGC genomics) or public repositories (HEK293: ATCC).

For siRNA transfections, the RNAiMAX reagent (Thermo Fisher Scientific) and the Opti-MEM medium (Thermo Fisher Scientific) were used following the manufacturers' instructions. A pool of siRNA against MYC (ON-TARGETplus SMARTpool MYC, L-003282-02-0020) and a control pool of siRNA (ON-TARGETplus Non-targeting Pool, D-001810-10-20) were purchased from GE Healthcare. Cells were harvested 48 hr after transfection.

### Expression and purification of recombinant MYC protein

The mRNA sequence of the dimerization and DNA-binding domain of human *MYC* was cloned into the pETM11 (EMBL-Heidelberg) vector. The protein was expressed as an N-terminally His-tagged protein in ArcticExpress (DE3)RIL cells (Agilent Technologies, Santa Clara, USA) after an OD of 0.8 was reached by induction with 0.1 mM isopropyl-β-thiogalactoside (IPTG) at 14°C for 18 hr. The protein was purified to homogeneity by metal affinity chromatography (Ni-NTA, Thermo Fisher Scientific) followed by anion exchange chromatography (AEC). AEC was performed using a MonoQ 10/100 GL column (GE Healthcare, Chicago, USA) equilibrated with 20 mM HEPES pH 8.0 and 0.25 M NaCl. The protein did not bind to the column but eluted in the flow through. Size exclusion chromatography was performed using a HiLoad 16/60 Superdex 200 prep grade column (GE Healthcare) to confirm its monomeric state and exclude aggregation. The protein was concentrated after AEC to 0.59 mg/ml based on the calculated extinction coefficient using ProtParam (SwissProt) and then flash frozen adding 10% glycerol for storage.

### Immunoblotting

Cells were either lysed in lysis buffer (50 mM HEPES pH 7.9, 140 mM NaCl, 1 mM EDTA, 1% Triton-X-100, 0.1% Sodium deoxycholate, 0.1% SDS) with protease and phosphatase inhibitor cocktails (Sigma-Aldrich) or harvested by trypsinization, counted and lysed directly in SDS sample buffer. The same number of cells or total protein amount was loaded for each sample on Bis-Tris or Tris-Glycine gels and transferred to PVDF membranes (EMD Millipore, Billerica, USA).

To detect the MYC protein the 9E10 antibody (amino acids: 410–419) or the anti-MYC antibody (Y69: N-terminus, Abcam, Cambridge, UK) were used. To detect Vinculin the V9131 anti-Vinculin antibody (Sigma-Aldrich) was used.

The immunoblots were visualized and quantified using the LAS-4000 mini (Fujifilm, Tokyo, Japan) or the Odyssey CLx Infrared Imaging System (LI-COR, Lincoln, USA).

The number of MYC molecules/cell was calculated by comparing the signal of recombinant MYC protein with cell lysates from U2OS cells in biological triplicates in quantitative immunoblots (Odyssey CLx Infrared Imaging System). To determine the nuclear MYC concentration, the size of U2OS nuclei was obtained from (*Koch et al., 2014*). Biological replicates are defined by individual U2OS cell harvestings.

### Chromatin immunoprecipitation (ChIP)

ChIP and ChIP-sequencing was performed as described previously (*Walz et al., 2014*). Briefly, cells were crosslinked with 1% formaldehyde for 10 min at room temperature. After extraction, nuclei were

lysed in lysis buffer (50 mM HEPES pH 7.9, 140 mM NaCl, 1 mM EDTA, 1% Triton-X-100, 0.1% Sodium deoxycholate, 0.1% SDS). The cell lysates were sonicated to reach an average DNA fragment size of 140-300 bp. Immunoprecipitation was performed 6 hours or overnight using Dynabeads (Thermo Fisher Scientific). For MYC and MNT immunoprecipitation, the anti-MYC antibody (N262, Santa Cruz, Dallas, USA) and the anti-MNT antibody (sc-769, Santa Cruz, Dallas, USA) were used, respectively. After washing, chromatin was eluted, de-crosslinked and treated with RNase A and Proteinase K. The DNA was purified by phenol/chloroform extraction and ethanol precipitation and analyzed by qPCR using MX3000P with Sybr green Mix (Thermo Fisher Scientific, ChIP) or sequenced on a Next-Seq500 after library preparation (NEBNext ChIP-Seq Library Prep Master Mix Set, NEB, Ipswich, USA).

## RT-qPCR

For RT-qPCR analysis, total cellular RNA was extracted using the peqGOLD TriFast (Peqlab, Erlangen, Germany) reagent. The first strand synthesis was performed using M-MLV Reverse Transcriptase (Thermo Fisher Scientific) and random hexamer primers (Roche, Basel, Switzerland). Results were normalized to β2-microglobulin expression. Primers are listed in *Supplementary file 3*. Technical replicates are defined by performing individual qPCR-analyses (CT-value estimation) with the same cDNA. Biological replicates are defined by individual U2OS cell harvestings.

## RNA-sequencing

For RNA-sequencing, total RNA was extracted using the RNeasy mini columns (Qiagen, Hilden, Germany) including on-column DNase I digestion. PolyA+-RNA was isolated from total RNA with the NEBNext Poly(A) mRNA Magnetic Isolation Module (NEB, Ipswich, USA). The NEBNext Ultra RNA Library Prep Kit for Illumina was used for library preparation following the manufacturer's instruction. Size selection of the libraries was performed with the AMPure XP Beads (Beckman Coulter, Brea, USA), followed by 12 PCR cycles for amplification. Library quantification and quality was assessed using the Experion Automated Electrophoresis System (Bio-Rad, Hercules, USA). The libraries were sequenced with the Illumina Genome Analyzer IIx (Illumina, San Diego, USA).

## Immunofluorescence

U2OS cells were treated with doxycycline/ethanol and were fixed with 4% paraformaldehyde. Fixed cells were treated with blocking solution (10% horse serum, 2% BSA and 5% sucrose in PBS) for 45 min after washing. Primary antibodies were incubated overnight at 4°C and incubated for 1 hr with the corresponding fluorescently labeled secondary antibodies at room temperature after washing (Tris-buffered saline with 0.1% Tween-20). Cells were mounted on object glass slides using aqua-fluoromount (Sigma) after washing and imaged under a confocal microscope (Nikon Ti-Eclipse). For fluorescent intensity measurement, ImageJ 1.50hr (*Schneider et al., 2012*) was used. Briefly, z stacked confocal images were converted to 12-bit images via maximum intensity projection and ROI assigned using ROI tool. The intensity was measured per cell nuclei for Hoechst and MYC. The fold change distribution of MYC was calculated by first taking MYC intensity normalized to Hoechst and then further normalizing the values to the median of the set of measurements. Finally $Log_2$ of the median normalized values were displayed as density plots on R using 'sm' library density function.

## Bioinformatic analyses

For RNA-sequencing, reads were aligned to the human reference genome (hg19) using BOWTIE v.0.12.8. The obtained BAM files were used for further analysis with R/BioConductor. For differential gene expression analysis EdgeR was employed. The rpkm values were calculated by mapping the read counts to the exons of the corresponding gene. Heat maps showing the gene expression changes were obtained using R.

ChIP-sequencing data were obtained from (*Walz et al., 2014*) and analyzed similarly. The MYC peaks were visualized using the wig files from MACS v.1.4.2 (*Feng et al., 2012*) and the Integrated Genome Browser software (*Nicol et al., 2009*). ChIP-sequencing samples from (*Walz et al., 2014*) were sequenced deeper for visualizing purposes.

Peak calling was performed with MACS v.1.4.2 (parameters: –keep-dup 3, variable p-value cut-off), SICER v.1.1 (parameters: redundancy 1, window 200, fragment size 90, effective genome fraction 0.74, gap 0, FDR 0.01) (*Zang et al., 2009*) and CCAT v.3.0 (parameters: default for transcription factors) (*Xu et al., 2010*). Overlapping peaks were determined with the 'intersectBed' function from BEDTools v.2.17.0 (*Quinlan, 2014*). Promoter regions are defined as –1.5 kb to +0.5 kb relative to the transcriptional start site if not indicated otherwise.

To analyze the occurrence of E-boxes in the MYC peak the CentriMo tool from the MEME Suite (*Bailey and Machanick, 2012*) was used. The empirical probability of consensus and non-consensus E-boxes was plotted with a moving average of 20 bp.

To calculate the distribution around the TSSs (heat maps) or the center of the MYC peak (occupancy), Seqminer v.1.3.3 was used. Heat maps were visualized using Treeview. MYC and H3K4me3 occupancies (in – and + Dox) were calculated using a window of $\pm$ 100 bp for the center of the MYC peak and used to obtain MYC recruitment. DNA binding of MYC and a mutant of MYC that is compromised in binding to WDR5, was analyzed similar: $5 \times 10^7$ reads were randomly chosen from fastq files (GSE60897), mapped with BOWTIE and reads in a window of $\pm$ 100 bp around the center of the MYC peak were counted using Seqminer. For Venn diagrams, publicly available peak annotations (GSE60897) were used and the overlap was calculated using Bedtools. For binned plots, genes were sorted and divided in equally sized-groups and the median or mean values are shown. As few genes show very high $EC_{50}$ values, sometimes one or two bins were not shown in the plots, for clarity.

$EC_{50}$ values for MYC binding at each promoter were calculated using the MYC nuclear concentrations and MYC occupancy in – and +Dox. MYC occupancy was calculated by subtracting the tag values of the input from the tag values of the MYC ChIP and resulting negative values were set to 0. The Michaelis-Menten model was used to perform the fitting of the data in GraphPad Prism and to calculate maximal occupancy of MYC and the $EC_{50}$ values.

For the estimation of the $EC_{50}$ values for regulation of MYC target genes, the MYC depletion RNA-sequencing experiment and the Dox titration RNA-sequencing experiment were used. The fold change of expression for each gene was calculated relative to siMYC. Only activated genes were considered. The $EC_{50}$ values were obtained using the fold change values and the nuclear MYC concentrations. Michaelis-Menten model was used to perform the fitting of the data in GraphPad Prism. To statistically test the difference of the distribution of the $EC_{50}$ values of different gene sets, the Mann-Whitney-Wilcoxon test was used. For functional analysis of the genes, Gene Set Enrichment Analyses (GSEA) were performed with the C5 collection of gene sets from the MSigDB (http://www.broadinstitute.org/gsea/msigdb) using the GSEAPreranked tool, where indicated. Gene ontology analysis was performed using DAVID (*Huang et al., 2009*) with default settings and gene ontology (GO) terms from the domains 'cellular component', 'molecular function' and 'biological process' as database. Sample size was not estimated by statistical methods but chosen based on common standards.

## Mathematical modeling

The temporal changes of the system's components are described by sets of ordinary differential equations (ODEs) and algebraic equations describing conservation relations. Model design, the particular equations and parameters are listed in the Appendix 1. An extended model analysis is provided in Appendix 2. Steady-state solutions were calculated by setting all time derivatives to zero and solving the resulting algebraic equation system numerically. All calculations were performed using Mathematica 10.2 (Wolfram Research, Inc.).

## Availability of supporting data

The data sets supporting the results of this article are available in the Gene Expression Omnibus repository, GEO: GSE77356. Data access: http://www.ncbi.nlm.nih.gov/geo/query/acc.cgi?acc=GSE77356

## Acknowledgements

We are grateful to Bruno Amati for sharing unpublished data. We thank Renate Metz, Angela Grün, Barbara Bauer and André Kutschke for excellent technical assistance and Laura Jänicke for her help in culturing non-transformed cell lines. We acknowledge all members of the ME/EW laboratory for helpful discussions and comments during the course of these experiments. This work

was supported by a grant from the Deutsche Forschungsgemeinschaft to EW (Emmy Noether Research Group, WO 2108/1-1) and to ME (Ei 222/12-1) and, the German Excellence Initiative via the Graduate School of Life Sciences (University of Würzburg) to EW and AB, the Studienstiftung des deutschen Volkes to LAJ as well as by the German Federal Ministry of Education and Research (BMBF) within the framework of the e:Med research and funding concept to ME (01ZX1307G) and JW (01ZX1307F).

## Additional information

### Funding

| Funder | Grant reference number | Author |
|---|---|---|
| Deutsche Forschungsgemeinschaft | Exzellenzinitiative des Bundes und der Länder Grant | Apoorva Baluapuri Elmar Wolf |
| Studienstiftung des Deutschen Volkes | | Lisa Anna Jung |
| Bundesministerium für Bildung und Forschung | 01ZX1307F | Jana Wolf |
| Bundesministerium für Bildung und Forschung | 01ZX1307G | Martin Eilers |
| Deutsche Forschungsgemeinschaft | Ei 222/12-1 | Martin Eilers |
| Deutsche Forschungsgemeinschaft | WO 2108/1-1 | Elmar Wolf |

The funders had no role in study design, data collection and interpretation, or the decision to submit the work for publication.

### Author contributions

FL, Carried out most of the experiments, Analyzed the data, Participated in the bioinformatics analyses, Helped to draft the manuscript; UB, Carried out the mathematical modeling, Helped to draft the manuscript, Analysis and interpretation of data; AB, Performed the immunofluorescence for MYC, Participated in the bioinformatics analyses, Helped preparing the quantitative immunoblots, Drafting or revising the article; SW, Performed the bioinformatical and statistical analyses, Drafting or revising the article; LAJ, Expressed and purified the recombinant MYC protein, Drafting or revising the article, Contributed unpublished essential data or reagents; BvE, Participated in the bioinformatical and statistical analyses, Drafting or revising the article; CK, Supervised the production of recombinant MYC protein, Analysis and interpretation of data, Drafting or revising the article, Contributed unpublished essential data or reagents; JW, Supervised the mathematical modeling, Analysis and interpretation of data, Drafting or revising the article; ME, Designed and coordinated the study, Drafted the manuscript, Analysis and interpretation of data; EW, Designed and coordinated the study, Supervised the experiments and drafted the manuscript, Analysis and interpretation of data

### Author ORCIDs

Elmar Wolf, http://orcid.org/0000-0002-5299-6335

## Additional files

### Supplementary files

• Supplementary file 1. This file provides the raw data and calculation for absolute protein quantification.

• Supplementary file 2. This file lists gene sets used in this study.

• Supplementary file 3. This file lists primer sequences used in this study.

## Major datasets

The following dataset was generated:

| Author(s) | Year | Dataset title | Dataset URL | Database, license, and accessibility information |
|---|---|---|---|---|
| Lorenzin F, Benary U, Jung LA, von Eyß B, Walz S, Kisker C, Wolf J, Eilers M, Wolf E | 2016 | Different promoter affinities account for specificity in MYC-dependent gene regulation | http://www.ncbi.nlm.nih.gov/geo/query/acc.cgi?acc=GSE77356 | Publicly available at NCBI Gene Expression Omnibus (accession no: GSE77356) |

The following previously published datasets were used:

| Author(s) | Year | Dataset title | Dataset URL | Database, license, and accessibility information |
|---|---|---|---|---|
| Thomas LR Wang Q, Grieb BC, Phan J, Foshage AM, Sun Q, Olejniczak ET, Clark T, Dey S, Lorey S, Alicie B, Howard GC, Cawthon B, Ess KC, Eischen CM, Zhao Z, Fesik SW, Tansey WP | 2015 | Interaction with WDR5 promotes target gene recognition and tumorigenesis by MYC | http://www.ncbi.nlm.nih.gov/geo/query/acc.cgi?acc=GSE60897 | Publicly available at NCBI Gene Expression Omnibus (accession no: GSE60897) |
| Walz S Lorenzin F, Morton J, Wiese KE, von Eyss B, Herold S, Rycak L, Dumay-Odelot H, Karim S, Bartkuhn M, Roels F, Wüstefeld T, Fischer M, Teichmann M, Zender L, Wei CL, Sansom O, Wolf E, Eilers M | 2014 | Activation and repression by oncogenic Myc shapes tumour-specific gene expression profiles | http://www.ncbi.nlm.nih.gov/geo/query/acc.cgi?acc=GSE44672 | Publicly available at NCBI Gene Expression Omnibus (accession no: GSE44672) |
| Sabò A Kress TR, Pelizzola M, de Pretis S, Gorski MM, Tesi A, Morelli MJ, Bora P, Doni M, Verrecchia A, Tonelli C, Fagà G, Bianchi V, Ronchi A, Low D, Müller H, Guccione E, Campaner S, Amati B | 2014 | Selective transcriptional regulation by Myc in cellular growth control and lymphomagenesis | http://www.ncbi.nlm.nih.gov/geo/query/acc.cgi?acc=GSE51008 | Publicly available at NCBI Gene Expression Omnibus (accession no: GSE51008) |

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

## Appendix 1

Appendix 1 provides the detailed description of the mathematical models 1 and 2. Schemes of models 1 and 2 are provided in *Appendix 1—figure 1*.

**Appendix 1—figure 1.** Schemes of models 1 and 2.

## Model 1

### Temporal change of MYC bound to DNA sequences

$$\frac{d[MYC@Ebox]}{dt} = v_1 - v_{1r} \tag{1}$$

$$\frac{d[MYC@NNNNNN]}{dt} = v_2 - v_{2r} \tag{2}$$

The hexamer sequence NNNNNN excludes the CACGTG sequence.

### Assumed conservation relations

$$[MYC_{total}] = [MYC] + [MYC@Ebox] + [MYC@NNNNNN] \tag{3}$$

$$[Ebox_{total}] = [Ebox_{unbound}] + [MYC@Ebox] \tag{4}$$

$$[NNNNNN_{total}] = [NNNNNN_{unbound}] + [MYC@NNNNNN] \tag{5}$$

### Rates

$$v_1 = k_1 \cdot [MYC] \cdot [Ebox_{unbound}] \tag{6}$$

$$v_{1r} = k_{-1} \cdot [MYC@Ebox] \tag{7}$$

$$v_2 = k_2 \cdot [MYC] \cdot [NNNNNN_{unbound}] \tag{8}$$

$$v_{2r} = k_{-2} \cdot [MYC@NNNNNN] \tag{9}$$

### Steady state assumption

$$\frac{d[MYC@Ebox]}{dt} = 0 \tag{10}$$

$$\frac{d[MYC@NNNNNN]}{dt} = 0 \tag{11}$$

The ratios of dissociation and association rate constant are substituted by the dissociation constants $K_{Ebox} = \frac{k_{-1}}{k_1}$ and $K_{NNNNNN} = \frac{k_{-2}}{k_2}$ .

## Model 2

### Temporal change of MYC bound to DNA sequences

$$\frac{d[MYC@Ebox]}{dt} = v_1 - v_{1r} \tag{12}$$

$$\frac{d[MYC@NNNNNN]}{dt} = v_2 - v_{2r} \tag{13}$$

$$\frac{d[MYC@EboxWDR5]}{dt} = v_3 - v_{3r} \tag{14}$$

$$\frac{d[MYC@NNNNNNWDR5]}{dt} = v_4 - v_{4r} \tag{15}$$

MYC can reversibly bind to four independent classes of DNA binding sites: CACGTG sequences (*Ebox*), unspecific DNA sequences (*NNNNNN*), WDR5 and CACGTG sequence simultaneously (*EboxWDR5*), as well as WDR5 and unspecific DNA sequences simultaneously (*NNNNNNWDR5*).

### Assumed conservation relations

$$\begin{aligned}[MYC_{total}] \quad &= [MYC] + [MYC@Ebox] + [MYC@NNNNNN] \\ &+ [MYC@EboxWDR5] + [MYC@NNNNNNWDR5] \end{aligned} \tag{16}$$

$$[Ebox_{total}] = [Ebox_{unbound}] + [MYC@Ebox] \tag{17}$$

$$[NNNNNN_{total}] = [NNNNNN_{unbound}] + [MYC@NNNNNN] \tag{18}$$

$$[EboxWDR5_{total}] = [EboxWDR5_{unbound}] + [MYC@EboxWDR5] \tag{19}$$

$$[NNNNNNWDR5_{total}] = [NNNNNNWDR5_{unbound}] + [MYC@NNNNNNWDR5] \tag{20}$$

The hexamer sequence NNNNNN excludes the CACGTG sequence. *Ebox_{total}* excludes CACGTG sequences in the vicinity of WDR5 (*EboxWDR5_{total}*).

### Rates

$$v_1 = k_1 \cdot [MYC] \cdot [Ebox_{unbound}] \tag{21}$$

$$v_{1r} = k_{-1} \cdot [MYC@Ebox] \tag{22}$$

$$v_2 = k_2 \cdot [MYC] \cdot [NNNNNN_{unbound}] \tag{23}$$

$$v_{2r} = k_{-2} \cdot [MYC@NNNNNN] \tag{24}$$

$$v_3 = k_3 \cdot [MYC] \cdot [EboxWDR5_{unbound}] \tag{25}$$

$$v_{3r} = k_{-3} \cdot [MYC@EboxWDR5] \tag{26}$$

$$v_4 = k_4 \cdot [MYC] \cdot [NNNNNNWDR5_{unbound}] \tag{27}$$

$$v_{4r} = k_{-4} \cdot [MYC@NNNNNNWDR5] \tag{28}$$

## Steady state assumption

$$\frac{d[MYC@Ebox]}{dt} = 0 \tag{29}$$

$$\frac{d[MYC@NNNNNN]}{dt} = 0 \tag{30}$$

$$\frac{d[MYC@EboxWDR5]}{dt} = 0 \tag{31}$$

$$\frac{d[MYC@NNNNNNWDR5]}{dt} = 0 \tag{32}$$

The ratios of dissociation and association rate constant are substituted by the dissociation constants $K_{Ebox} = \frac{k_{-1}}{k_1}$, $K_{NNNNNN} = \frac{k_{-2}}{k_2}$, $K_{EboxWDR5} = \frac{k_{-3}}{k_3}$, and $K_{NNNNNNWDR5} = \frac{k_{-4}}{k_4}$.

# Definition of occupancy

Occupancy of E-boxes is defined by the proportion of E-boxes bound by MYC in total number of E-boxes in percentage at steady state:

$$occupancy_{Ebox} = \frac{[MYC@Ebox]}{[Ebox_{total}]} \cdot 100\%.$$

Likewise, occupancy of NNNNNN sequences, of Ebox-WDR5 and NNNNNN-WDR5 binding sites are given by

$occupancy_{NNNNNN} = \frac{[MYC@NNNNNN]}{[NNNNNN_{total}]} \cdot 100\%$ ,

$occupancy_{EboxWDR5} = \frac{[MYC@EboxWDR5]}{[EboxWDR5_{total}]} \cdot 100\%$ , and

$occupancy_{NNNNNNWDR5} = \frac{[MYC@NNNNNNWDR5]}{[NNNNNNWDR5_{total}]} \cdot 100\%$ , respectively.

## Remark on steady state assumption

Our modelling approach partially builds on data of ChIP-sequencing experiments. ChIP-sequencing experiments are based on the general notion that the number of DNA-bound molecules (tags) presumably does not change (at least) for the time frame of conducting the experiment. We therefore use the steady state assumption for calculating apparent dissociation constants, which imply ratios on dissociation and association rate constants.

## Parameters

The amount of $MYC_{total}$ is varied up to $4.6 \cdot 10^6$ molecules, which is the maximal amount experimentally measured in U2OS cells under different treatments (EtOH, Dox, and siRNA; see *Supplementary file 1*).

The following tables (*Appendix 1—table 1*, *Appendix 1—table 2* and *Appendix 1—table 3*) list all parameters used in model 1 and 2.

**Appendix 1—table 1.** Parameters generally applicable to models 1 and 2.

| Parameter | Value | Unit | Comment, Reference |
|---|---|---|---|
| *Vol* | $7.7 \cdot 10^{-13}$ | L | 1) |
| $Genome_{total}$ | $9.5 \cdot 10^9$ | base pairs | 2) |

**Appendix 1—table 2.** Parameters of model 1.

| Parameter | Value | Unit | Comment, Reference |
|---|---|---|---|
| $Ebox_{total}$ | $9.3 \cdot 10^5$ | molecules | 3) |
| $NNNNNN_{total}$ | $9.5 \cdot 10^9$ | molecules | 4) |
| $K_{Ebox}$ | $4.6 \cdot 10^1$ | molecules | 5); 0.1 nM (*Guo et al., 2014*) |
| $K_{NNNNNN}$ | $9.3 \cdot 10^3$ | molecules | 5); 20 nM (*Guo et al., 2014*) |

**Appendix 1—table 3.** Parameters of model 2.

| Parameter | Value | Unit | Comment, Reference |
|---|---|---|---|
| $Ebox_{total}$ | $9.2 \cdot 10^5$ | molecules | 6) |
| $NNNNNN_{total}$ | $9.5 \cdot 10^9$ | molecules | 6) |
| $EboxWDR5_{total}$ | $1.2 \cdot 10^4$ | molecules | 6) |
| $NNNNNNWDR5_{total}$ | $4.7 \cdot 10^4$ | molecules | 6) |
| $K_{Ebox}$ | $4.6 \cdot 10^1$ | molecules | 5); 0.1 nM (*Guo et al., 2014*) |
| $K_{NNNNNN}$ | $9.3 \cdot 10^3$ | molecules | 5); 20 nM (*Guo et al., 2014*) |
| $K_{EboxWDR5}$ | $4.3 \cdot 10^{-4}$ | molecules | 7) |
| $K_{NNNNNNWDR5}$ | $8.6 \cdot 10^{-2}$ | molecules | 7) |

## Comments

1. Walz, Lorenzin *et al.* show that Dox-induced MYC is predominantly localized in U2OS cell nuclei ([*Walz et al., 2014*], extended data Figure 1B). We therefore consider only the nuclear volume of an U2OS cell. The nucleus dimensions were quantified as approximately 16 μm in width, 23 μm in length, and 4 μm in height (*Koch et al., 2014*). We assumed an ellipsoid to calculate the volume of an U2OS cell nucleus.

2. $Genome_{total}$ is the number of considered base pairs estimated for the female hypertriploid karyotype of U2OS cells. Total base pairs are counted in the haploid unmasked human reference genome hg19 without taking the Y chromosome into account and excluding 'N' and

'chrN_random-sequences'. The number is multiplied by 2 to account for diploid cells and subsequently multiplied by a factor of 1.7 (*Ben-Shoshan et al., 2014*) to account for the hypertriploid karyotype of U2OS cells (http://www.lgcstandards-atcc.org/products/all/HTB-96.aspx#characteristics) (*Janssen and Medema, 2013*).

3. $Ebox_{total}$ is calculated by counting CACGTG sequences in the human reference genome hg19 without taking the Y chromosome into account and excluding 'N' and 'chrN_random-sequences'. The number is multiplied by 2 and 1.7 to account for the karyotype of U2OS cells.

4. The total number of possible hexameric sequences ($NNNNNN_{total}$) in the genome of U2OS cells is estimated by subtracting $Ebox_{total}$ from $Genome_{total}$.

5. To convert the published values of dissociation constants given in [nM] and [μM] into [molecules] the average volume of a single U2OS cell nucleus listed in *Appendix 1—table 1* is used.

6. To obtain $Ebox_{total}$, $NNNNNN_{total}$, $EboxWDR5_{total}$, and $NNNNNNWDR5_{total}$, we first downloaded processed data of a WDR5-ChIPseq experiment (*Thomas et al., 2015*) from Gene Expression Omnibus database (GSE60897), which comprised 17,171 WDR5 peaks. Next, we identified 27,853 unique RefSeq-TSS (USCS, hg19.refGene), which enumerated to 22,346 promoters assuming −1.5 kb to +0.5kb as promoter region and joining overlapping promoter regions. 3,408 of these 22,346 promoters contain WDR5 as well as a CACGTG sequence. This number (3,408) is multiplied by 2 and 1.7 to account for the karyotype of U2OS cells yielding $EboxWDR5_{total}$. $Ebox_{total}$ is given by subtracting $EboxWDR5_{total}$ from $Ebox_{total}$ listed in *Appendix 1—table 2*. $NNNNNNWDR5_{total}$ is calculated by subtracting $EboxWDR5_{total}$ from all called WDR5 peaks (i.e., 17,171 WDR5 peaks multiplied by 2 and 1.7 to account for U2OS cell karyotype). $NNNNNN_{total}$ results from subtracting the sum of $Ebox_{total}$, $EboxWDR5_{total}$, and $NNNNNNWDR5_{total}$ from $Genome_{total}$.

7. The dissociation constant of MYC and EboxWDR5 is estimated by multiplying the individual dissociation constants of MYC and WDR5 ($9.3 \cdot 10^{-6}$ M; [*Thomas et al., 2015*]) as well as MYC and E-boxes ($0.1 \cdot 10^{-9}$ M; [*Guo et al., 2014*]). This approach is based on our presumption that under constant pressure the standard change of Gibbs free energy at equilibrium of reversible binding of MYC to EboxWDR5 is equal to the sum of the standard change of Gibbs free energies at equilibria of reversible MYC-WDR5-association and reversible MYC-CACGTG-association. Similarly, the dissociation constant of MYC and NNNNNNWDR5 is estimated by multiplying the individual dissociation constants of MYC and WDR5 ($9.3 \cdot 10^{6}$ M; (*Thomas et al., 2015*]) as well as MYC and unspecific DNA ($20 \cdot 10^{9}$ M; [*Guo et al., 2014*]).

# Appendix 2

Appendix 2 provides an extended analysis and discussion of the modelling approach.

## Motivation and aims

The mathematical models introduced in the manuscript consider two kinds of parameters: dissociation constants ($K_{Ebox}$, $K_{NNNNNN}$, $K_{EboxWDR5}$, and $K_{NNNNNNWDR5}$) and total numbers of molecules and binding sites ($[MYC_{total}]$, $[Ebox_{total}]$, $[NNNNNN_{total}]$, $[EboxWDR5_{total}]$, and $[NNNNNNWDR5_{total}]$). In Appendix 1, we provide a particular set of parameter values for each model. Here, in Appendix 2, we use model 1 to comprehensively dissect the impact of model parameters on the occupancy of E-boxes by MYC. We investigate:

1. how the occupancy of E-boxes by MYC depends on the total number of MYC molecules allowing for a wide range of dissociation constants $K_{NNNNNN}$ and $K_{Ebox}$, and

2. to what extend this dependency is affected by variations of E-box and NNNNNN sequence counts to estimate the impact of heterochromatin or altered karyotype.

## Summary of results

Our comprehensive analysis reveals that a higher affinity of MYC to E-boxes than NNNNNN sequences is sufficient to observe higher occupancy of E-boxes than of NNNNNN sequences by MYC in the measured range of MYC abundance. Second, we demonstrate that the dependence of E-boxes occupancy and NNNNNN sequence occupancy on MYC in the measured range of MYC abundance is strongly influenced by the ratio of the respective dissociation constants. Finally, we show that a variation of the number of E-boxes and NNNNNN sequences to account for heterochromatin or deviation of karyotype hardly affects our results.

## Discussion of the parameters of model 1

As mentioned above, five parameters characterize model 1: $K_{Ebox}$, $K_{NNNNNN}$, $[Ebox_{total}]$, $[NNNNNN_{total}]$, and $[MYC_{total}]$. Since the values of $[Ebox_{total}]$ and $[NNNNNN_{total}]$ are relatively defined for a given genome, we will initially fix $[Ebox_{total}]$ to $9.3 \cdot 10^5$ and $[NNNNNN_{total}]$ to $9.5 \cdot 10^9$ in our analysis to reduce parameter space. These values are calculated from human reference genome (hg19) assuming the karyotype of U2OS cells (see comments in Appendix 1). In a second step, we will extend our analysis to investigate the impact of other values of total number of E-boxes and NNNNNN sequences to take heterochromatin and karyotype alterations into account.

The values of $[MYC_{total}]$ were experimentally determined to range from $7.1 \cdot 10^3$ to $4.6 \cdot 10^6$ molecules (**Supplementary file 1**). Therefore, $[MYC_{total}]$ is varied in our analyses over orders of magnitude on a logarithmic scale from $3 \cdot 10^3$ to molecules.

The values of the dissociation constants $K_{NNNNNN}$ and $K_{Ebox}$ are varied over a very wide range (from $10^{-7}$ to $10^7$ molecules) in this analysis to cover all possible conditions.

## Results of the model analysis

### Conditions for higher occupancy of E-boxes than of NNNNNN sequences

*Appendix 2–figure 1* illustrates regions of certain occupancy in the three dimensional parameter space of dissociation constants $K_{NNNNNN}$ and $K_{Ebox}$ as well as total amount of MYC $[MYC_{total}]$. It shows that higher occupancy of E-boxes (*Appendix 2–figure 1A*) than of NNNNNN sequences (*Appendix 2–figure 1B*) is observable for most combinations of dissociation constants $K_{NNNNNN}$ and $K_{Ebox}$ in the considered range of total amount of MYC. For instance, occupancy of E-boxes can reach 90% at $[MYC_{total}]$ values in the order of $10^6$ molecules. In contrast, the occupancy of NNNNNN sequences is generally low in the order of 0.1% or less. This is mainly due to the vast excess of NNNNNN sequences compared to E-boxes.

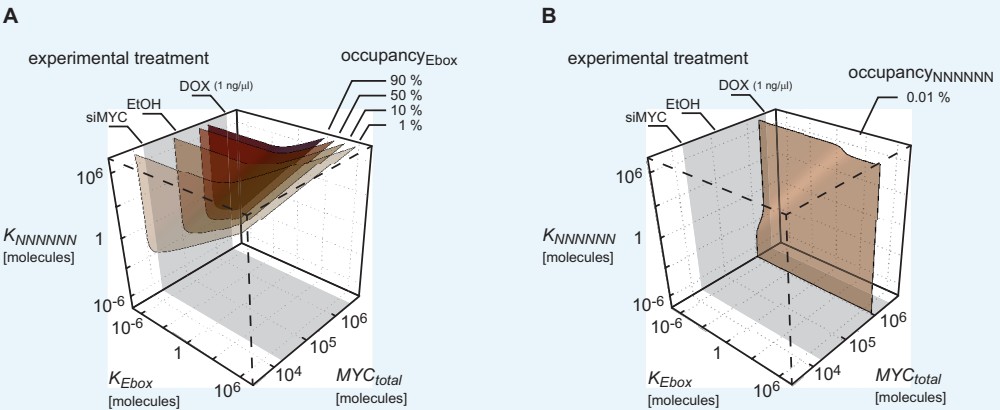

**Appendix 2–figure 1.** Plot shows surfaces of equal occupancy as function of dissociation constants $K_{NNNNNN}$ and $K_{Ebox}$ as well as total amount of MYC. Different regions of occupancy of E-boxes (**A**) and NNNNNN sequences (**B**) are shown in this parameter space. The experimentally measured range of the total amount of MYC is marked in grey. Total MYC concentrations of U2OS cells were changed by different experimental treatments as indicated on the upper axis. To simulate U2OS cells, $[Ebox_{total}] = 9.3 \cdot 10^5$ and $[NNNNNN_{total}] = 9.5 \cdot 10^9$ are assumed.

*Appendix 2–figure 2* shows the parameter space of *Appendix 2–figure 1A* from two different perspectives, i.e. rotated into the $[MYC_{total}]$-$K_{Ebox}$ -plane and into the $K_{Ebox}$-$K_{NNNNNN}$-plane. For simplification only the surface of 50% occupancy of E-boxes by MYC is shown. Looking from these two perspectives, it seems that a critical total amount of MYC is necessary to observe occupancies above 50% and the dissociation constant $K_{NNNNNN}$ must be larger than the dissociation constant $K_{Ebox}$. Indeed, an analytical analysis proofs (see below) that $K_{NNNNNN} > K_{Ebox}$ is both a necessary and a sufficient condition to observe $occupancy_{Ebox} > occupancy_{NNNNNN}$. This implies that even in the case of small differences in the dissociation constants of MYC to E-boxes and NNNNNN sequences, differential occupancies occur.

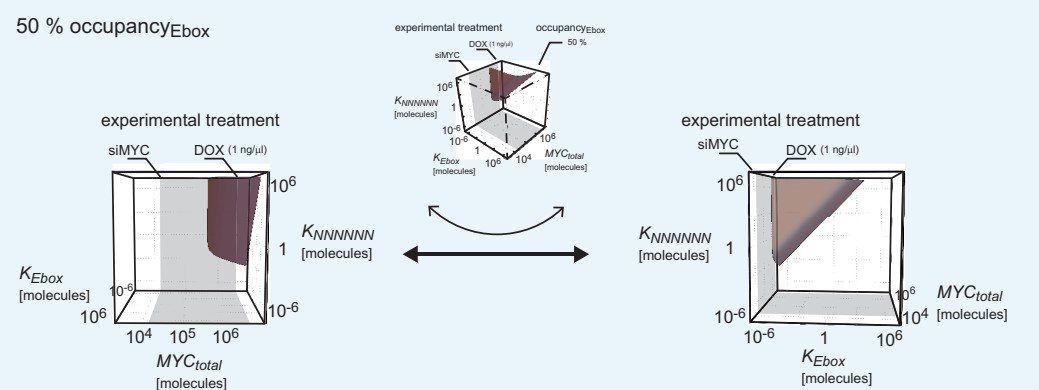

**Appendix 2–figure 2.** Plot shows surface of $occupancy_{Ebox} = 50\%$ of *Appendix 2–figure 1A* in two different orientations: rotated into the $[MYC_{total}]$-$K_{Ebox}$ -plane and into the $K_{Ebox}$-$K_{NNNNNN}$ -plane.

## The ratio of both dissociation constants impacts the difference between occupancies of E-boxes and NNNNNN sequences

Next, we investigate how strongly the occupancies of E-boxes and NNNNNN sequences can differ. To explore whether absolute values or ratios of the dissociation constants $K_{NNNNNN}$ and $K_{Ebox}$ play an important role to separate the two occupancy curves, we vary the value of $K_{Ebox}$ as well as the ratio of $K_{NNNNNN}$ to $K_{Ebox}$ in a systematic fashion (*Appendix 2–figure 3*). We observe that for a given value of $K_{Ebox}$, the difference between the occupancy of E-boxes and NNNNNN sequences is the greater, the greater the ratio of $K_{NNNNNN}$ to $K_{Ebox}$ . Moreover, for a given ratio of the dissociation constants, the value of $K_{Ebox}$ hardly impacts the occupancy curves. This demonstrates that the ratio of the dissociation constants but not their absolute values predominantly determines whether distinct occupancies of E-boxes and NNNNNN sequences can be observed. Please note that the changes of $K_{Ebox}$ in the calculations for *Figure 3E* simultaneously changed the ratio of $K_{NNNNNN}$ to $K_{Ebox}$ and therefore affected the EC50 value.

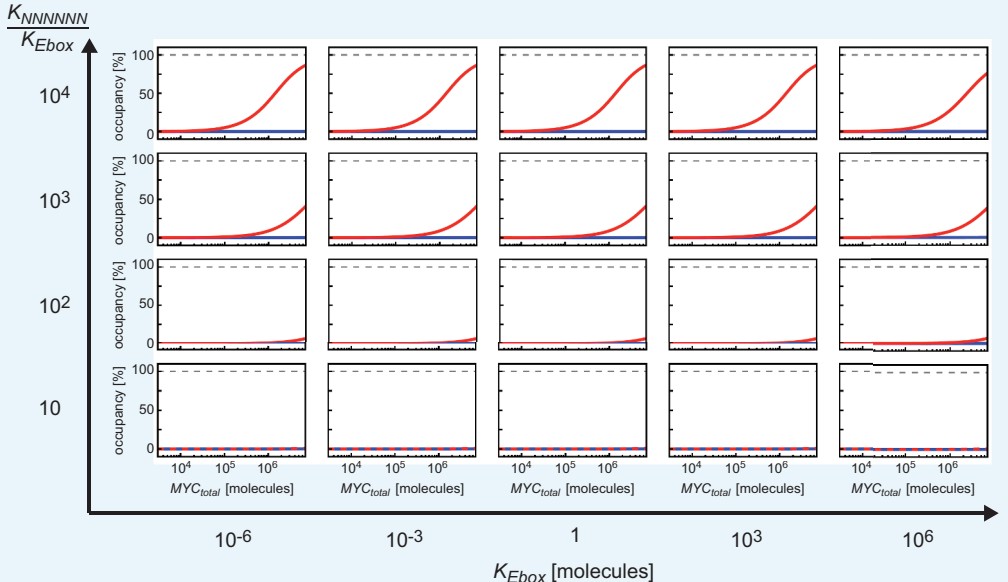

**Appendix 2–figure 3.** Ratio of the dissociation constants $K_{NNNNNN}$ and $K_{Ebox}$ determines how much occupancy of E-boxes differs from occupancy of NNNNNN sequences. The panels show the dependency of the occupancies of E-boxes (red curve) and NNNNNN sequences (blue curves) on total number of MYC molecules. For each panel a different combination of

dissociation constants $K_{NNNNNN}$ and $K_{Ebox}$ are chosen. The panels are displayed in a matrix sorted for increasing values of $K_{Ebox}$ and increasing ratios $K_{NNNNNN}$ to $K_{Ebox}$.

## Variations of number of E-boxes and NNNNNN sequences hardly affect the previous results

So far, a fixed number of E-boxes and NNNNNN sequences was considered in the simulations ($[Ebox_{total}] = 9.3 \cdot 10^5$ and $[NNNNNN_{total}] = 9.5 \cdot 10^9$). Taking into account that the euchromatic sequence makes up approximately 90% of the human genome (International Human Genome Sequencing Consortium, 2004) and the karyotype of U2OS cells substantially differs from that of normal human cells (**Ben-Shoshan et al., 2014**), the impact of the number of E-boxes and NNNNNN sequences on occupancy of E-boxes is now investigated. The analysis shows that a change in the numbers of E-boxes and NNNNNN sequences hardly affects the occupancy of E-boxes with respect to total MYC (**Appendix 2–figure 4**).

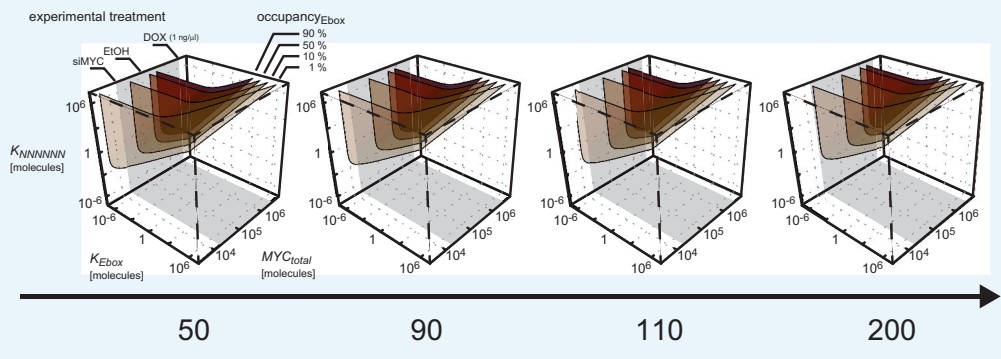

**Appendix 2–figure 4.** Impact of number of E-boxes and NNNNNN sequences. Different combinations of number of E-boxes and NNNNNN sequences are assumed in the simulations: $[Ebox_{total}] = 9.3 \cdot 10^5$ and $[NNNNNN_{total}] = 9.5 \cdot 10^9$ represent the reference values and the range of 50% to 200% of these reference values is taken into account. Surfaces of equal occupancy of E-boxes are shown as function of dissociation constants $K_{NNNNNN}$ and $K_{Ebox}$ as well as total amount of MYC.

## Proof

Using equations 1–11 (Appendix 1), we derive the following equation system:

$$\left| \begin{matrix} 0 = [MYC] \cdot (1 - occupancy_{Ebox}) - K_{Ebox} \cdot occupancy_{Ebox} \\ 0 = [MYC] \cdot (1 - occupancy_{NNNNNN}) - K_{NNNNNN} \cdot occupancy_{NNNNNN} \end{matrix} \right|$$

This yields expressions of $K_{NNNNNN}$ and $K_{Ebox}$:

$$K_{Ebox} = [MYC] \cdot \frac{(1 - occupancy_{Ebox})}{occupancy_{Ebox}}$$
$$K_{NNNNNN} = [MYC] \cdot \frac{(1 - occupancy_{NNNNNN})}{occupancy_{NNNNNN}}$$

Assuming $K_{NNNNNN} > K_{Ebox}$ holds, then

$$\Rightarrow \frac{(1 - occupancy_{NNNNNN})}{occupancy_{NNNNNN}} > \frac{(1 - occupancy_{Ebox})}{occupancy_{Ebox}}$$

$$\Rightarrow occupancy_{Ebox} > occupancy_{NNNNNN}$$

Similarly it can be proven that if $occupancy_{Ebox} > occupancy_{NNNNNN}$ then $K_{NNNNNN} > K_{Ebox}$ holds.

