## [Decision Letter]

Thank you for submitting your article "Different promoter affinities account for specificity in MYC-dependent gene regulation" for consideration by *eLife*. Your article has been favorably evaluated by Jessica Tyler (Senior editor) and three reviewers, one of whom, Chi Dang, is a member of our Board of Reviewing Editors. The following individual involved in review of your submission has agreed to reveal their identity: David Levens (peer reviewer).

The reviewers have discussed the reviews with one another and the Reviewing Editor has drafted this decision to help you prepare a revised submission. The requested revision mostly requires additional analysis of extant data and does not necessarily require new experiments. For major point #1, it is possible that you may have the data already as alluded to in the Walz et al. paper (Nature 2014; extended Figure 1 immunofluorescent microscopy study).

Summary:

The manuscript by Lorenzin et al. addresses the conundrum of widespread MYC binding to the genome and its specific regulation of various cell biological functions in the context of its general gene activation function, seemingly without any specificity. By measuring MYC DNA binding affinities, its association with WDR5, the authors were able to stratify promoters by change in MYC occupancy. The data indicate that low-affinity promoters show the greatest increase in MYC recruitment at high levels of MYC whereas the high affinity promoters show very little change over the same range. This result is similar to that of Fernandez et al. (Fernandez PC Genes & Dev 2003) that employed large-scale ChIP to propose MYC occupancy dependance by promoter affinity. Using mathematical modeling, Lorenzin et al. are able to assign a binding affinity to each promoter (EC50) and make the intriguing conclusion that E box binding alone is not strong enough to explain how MYC is tethered to the majority of its DNA targets. Finally, the authors measure gene expression by titrating the level of MYC which, consistent with their model, reveals a correlation between promoter affinity and the level to which expression from these genes was increased. Specifically, they demonstrate that high-affinity binding sites are almost fully occupied by MYC and could not be further bound, resulting in a seemingly lack of gene activation by ectopic MYC. This is best illustrated in an artificial system of U2OS cells that have relatively low MYC levels despite being a cancer cell line. Upon activation of ectopic MYC, genes encoding ribosome biogenesis genes, which have high-affinity MYC binding sites, could not be further bound and hence do not seem to be activated. However, knockdown of endogenous MYC in U2OS cells resulted in diminished binding of MYC and expression of ribosome biogenesis genes. By monitoring MXD at E-boxes, the authors show an in vivo competition with MYC that is sensitive to MYC levels. The authors also show that MIZ repression explains most of the negative gene regulation with MYC that is again dependent on the level of MYC binding. Overexpression of MYC, on the other hand, results in MYC binding to lower affinity sites, resulting in ectopic activation of genes such as those involved in angiogenesis and suppression of cell adhesion genes. The authors also use publicly available data for the MYC-mediated mouse lymphoma and also documented MYC's role in inducing ribosome biogenesis genes (which was previously documented by Ji et al. PLoS One 2011). Using data deposited on WDR5, the authors corroborate the role of WDR5 in recruiting MYC to DNA for gene regulation. In addition, the experimental data, the authors also provide mathematical modeling, providing further insights into how MYC binding affinities affect gene expression and drive specific biological functions.

Overall, this body of work is compelling, although the authors should openly acknowledge that the U2OS cell line is complicated by its intolerance of high expression of MYC (documented in Walz et al. Nature 2014; extended Figure 1), resulting in cell death upon prolonged exposure to high MYC levels (>3days). This nuance could well skew MYC's ectopic activities that are specific for this cell line and hence not generalizable to cancer cells driven by high MYC such as Burkitt's lymphoma or neuroblastoma. For example, the authors should compare the 'ectopic' MYC targets among the different datasets (U2OS, Emu-MYC normal-pretumor-tumor samples, and other datasets referenced in the manuscript). Are there cell-type-independent MYC ectopic target genes similar to cell-type-independent core MYC targets that are involved in ribosome biogenesis?

Essential revisions:

Most of the conclusion presented in the manuscript i.e., the EC50 values assigned to promoter and the necessity of secondary contacts of MYC to DNA via protein-protein interactions derive from the author's seminal experiments to determine the number of MYC molecules per cell. If these initial numbers are wrong, then the values used for calculations in subsequent experiments are dubious. The authors should address the following points, that if dealt with would increase the significance and rigor of the work.

1) The authors use recombinant, purified protein to calibrate immunoblots in order to calculate the absolute number of MYC molecules per cell using cell lysate. This method however will only indicate the average level of MYC for the entire population. It remains possible that some cells express no MYC while others express levels of MYC far exceeding the average. The authors should employ immunofluorescence or a similar technique to demonstrate MYC expression in their system is relatively homogenous.

2) In Figure 1, the authors define the amount of MYC in their U2OS cells -/+ Dox. It appears that the amount of MYC in ETOH treated cells is beyond the range of their lowest standard (compare Figure 1 to Figure 2). Indeed, if the fold range of expression is truly 14-fold as presented in the manuscript, then standards covering only a four-fold range (2.4 ng to 9.6 ng in Figure 2) is insufficient. The authors should repeat the experiment with an expanded range of standards to insure that they have assigned the correct number of MYC molecules to the 'control' condition.

3) The author's analysis of transcription upon knock-down of MYC revealed that high-affinity promoters showed the greatest change in gene expression upon MYC depletion. This finding seems to be in disagreement with the finding from Nie et al. (Nie Z, 2012) in which B cells expressing very low levels of MYC show a proportional decrease of MYC binding to all genomic targets. Can the authors resolve this discrepancy? At the very least, the authors should perform ChIPseq upon MYC knock-down in their U2OS model and determine promoter occupancy to complement their gene expression experiments.

4) There appears to be somewhat of a disconnect between the accounting of MYC binding at promoters and the heat map shown in Figure 1. From the heat map across all genes it appears that MYC follows a monotonic binding pattern all the way to the bottom (I don't see a negative region in the heat map) – in contrast the accounting of MYC positive versus negative promoters is highly dependent on the selection of p-values and/or false discovery rates to create an arbitrary distinction between + and –. Some discussion of this may be warranted. Perhaps the negative peaks simply reflect a continuum of MYC binding as it slides below the level of the technical background.

5) There are several issues related to the model. The authors only model the influence of WDR in concert with E-boxes; what is the level of binding and the influence of MYC at WDR^+^, E-box^-^ promoters? This class should be considered, too. What is the extent of MYC binding, expression levels and functional GSEA for this class of promoters?

6) The model assumes equilibrium (or at least steady state) binding. While this is often a reasonable assumption, at promoters, in vivo, there are many energy-dependent and kinetic steps, that might skew that apparent on and off rates. For example, the authors' own previous work showed that MYC degradation during the transcription-cycle might be required to increase expression. Such degradation will increase the apparent MYC off-rate, artificially depressing apparent affinity of MYC for the promoter. The kinetic limitations of the model should be discussed in the text or in the supplemental material.

7) Some discussion of expression levels for the various functional classes of MYC-target promoters is warranted. It has previously been suggested that MYC binding is related to output levels. It may be that some cellular processes (such as translation) demand high output, and that it is the high level of transcription at these genes that licenses MYC binding, that in turn stimulates more transcription. Accompanying Tables of genes in groups identified in the Figures should be provided. (e.g., ribosome, extracellular matrix, solute transporter etc.). Further, the authors should provide an analysis of selected metabolic pathways, such as glycolysis and glutaminolysis, that may be regulated in U2OS cells.

---

## [Author Response]

*Overall, this body of work is compelling, although the authors should openly acknowledge that the U2OS cell line is complicated by its intolerance of high expression of MYC (documented in Walz et al. Nature 2014; extended Figure 1), resulting in cell death upon prolonged exposure to high MYC levels (>3days).*

We agree and now state: “Notably, prolonged exposure (>3 days) to doxycycline and hence long-term ectopic expression of MYC induces apoptosis in U2OS cells (Walz et al., 2014); therefore all subsequent analyses were performed 28-30 hours after addition of doxycycline.” at the beginning of the Results section.

This nuance could well skew MYC's ectopic activities that are specific for this cell line and hence not generalizable to cancer cells driven by high MYC such as Burkitt's lymphoma or neuroblastoma. For example, the authors should compare the 'ectopic' MYC targets among the different datasets (U2OS, Emu-MYC normal-pretumor-tumor samples, and other datasets referenced in the manuscript). Are there cell-type-independent MYC ectopic target genes similar to cell-type-independent core MYC targets that are involved in ribosome biogenesis?

In reply to this comment, we compared the functional groups of low-affinity target genes defined in the U2OS cell system with those the Eµ-Myc model and describe the data in the new panels H and I of Figure 5—figure supplement 1. Indeed, several functional classes of proteins encoded by low affinity genes, like “transmembrane transporters” are also regulated during lymphomagenesis (subsection “Binding affinity determines MYC-mediated transcriptional responses at different MYC concentrations”, first paragraph).

Essential revisions:

*1) The authors use recombinant, purified protein to calibrate immunoblots in order to calculate the absolute number of MYC molecules per cell using cell lysate. This method however will only indicate the average level of MYC for the entire population. It remains possible that some cells express no MYC while others express levels of MYC far exceeding the average. The authors should employ immunofluorescence or a similar technique to demonstrate MYC expression in their system is relatively homogenous.*

To address the cell-to-cell variance of MYC levels, we performed immunofluorescence of MYC with three different antibodies at physiological and ectopic MYC concentrations. The new Figure 2—figure supplement 2 shows examples and quantification of the immunofluorescence. Of course, there is some variation of MYC levels between cells, but 80% of U2OS cell population differs less than +/– 3.7 fold for endogenous and +/– 2.9 fold for overexpressed MYC. To estimate the impact of this variation, we tested the effect of this maximal variation (80% of the U2OS cells) in the mathematical model. The results are shown as yellow shaded areas in Figure 2—figure supplement 2. Importantly, high-affinity binding sites (characterized by E-boxes and WDR5 binding) are predicted to be highly occupied for MYC even if protein levels of endogenous MYC are 3.6 fold lower than on average. The new data is described (subsection “Absolute quantification of nuclear MYC allows an estimate of MYC binding affinities”, first paragraph) and interpreted (subsection “Binding to DNA and to WDR5 accounts for high promoter affinity”, third paragraph) in the Results section.

*2) In Figure 1, the authors define the amount of MYC in their U2OS cells −/+ Dox. It appears that the amount of MYC in ETOH treated cells is beyond the range of their lowest standard (compare Figure 1 to Figure 2). Indeed, if the fold range of expression is truly 14-fold as presented in the manuscript, then standards covering only a four-fold range (2.4 ng to 9.6 ng in Figure 2) is insufficient. The authors should repeat the experiment with an expanded range of standards to insure that they have assigned the correct number of MYC molecules to the 'control' condition.*

In the former version of the manuscript we quantified a small range of defined amounts of recombinant MYC protein. We calibrated the cellular MYC concentration in cells treated with a concentration of doxycycline, which leads to a signal close to that of the recombinant protein. We then estimated concentrations in all other conditions based on relative changes and assumed linearity (fluorescence based immunoblots, Licor).

We agree with the reviewers that an absolute estimation of MYC levels at each condition (siMYC, EtOH, Dox-titration) is more reliable. We therefore repeated the experiment using an extended range of standards covering the signal intensity from all biological conditions analyzed in biological triplicates and calculated a fitting model for the titration of recombinant MYC protein (which indeed does not behave linear). This regression analysis was used to estimate cellular MYC concentrations. The numbers are very similar for low cellular MYC concentrations (siMYC, EtOH) and differ by a maximal factor of three for the highest doxycycline concentration if compared with the former analysis. The new blots are shown in Figure 2—figure supplement 1 and the calculations are presented in [Supplementary-material SD1-data].

*3) The author's analysis of transcription upon knock-down of MYC revealed that high-affinity promoters showed the greatest change in gene expression upon MYC depletion. This finding seems to be in disagreement with the finding from Nie et al. (Nie Z, 2012) in which B cells expressing very low levels of MYC show a proportional decrease of MYC binding to all genomic targets. Can the authors resolve this discrepancy? At the very least, the authors should perform ChIPseq upon MYC knock-down in their U2OS model and determine promoter occupancy to complement their gene expression experiments.*

To address this comment, we performed a new MYC ChIP-sequencing experiment upon depletion of MYC by siRNA and a side-by-side ChIP-sequencing experiment with control cells (siCtr). The data is presented in the new Figure 5—figure supplement 1. We observed that (i) endogenous MYC binds to thousands of promoters (siCtr), confirming our previous data (ii) MYC binding is not homogenous but greatly differs for low- and high-affinity binding sites, (iii) binding is abolished (or at least strongly decreased) at the majority of the binding sites, including high-affinity binding sites like the promoters of ribosomal protein genes. Strikingly, the RNA-sequencing experiment and the new ChIP-sequencing experiment upon MYC depletion nicely correlate since the genes that are most strongly affected in expression also show the strongest decrease in MYC binding to their promoters (based on fold change, see Figure 5—figure supplement 1). The new data is described (subsection “Binding affinity determines MYC-mediated transcriptional responses at different MYC concentrations”, first paragraph) in the Results section.

We think that our data fit well to those reported by Nie et al. For instance, in Figure S2F of Nie et al. (Cell, 2012), MYC binding in T4 (after 4 hours of activation) is not uniform across all genes and MYC binding further increases at T14 on some promoters that are weakly bound at T4 (e.g. group “III”, “VII”), but appears to be saturated at T4 at high affinity binding sites (e.g. group “I”).

*4) There appears to be somewhat of a disconnect between the accounting of MYC binding at promoters and the heat map shown in Figure 1. From the heat map across all genes it appears that MYC follows a monotonic binding pattern all the way to the bottom (I don't see a negative region in the heat map) – in contrast the accounting of MYC positive versus negative promoters is highly dependent on the selection of p-values and/or false discovery rates to create an arbitrary distinction between + and –. Some discussion of this may be warranted. Perhaps the negative peaks simply reflect a continuum of MYC binding as it slides below the level of the technical background.*

To address these comments, we include a new supplemental Figure (Figure 1—figure supplement 1), in which we show the analysis with different peak callers and parameters. Importantly, the reported number of MYC binding sites appears to be very robust, as peak calling with different programs using default parameters results in very similar called peaks and peak numbers (Figure 1—figure supplement 1). Panel C illustrates how negative peaks are increasing when decreasing the p-value adjustment in line with increasing positive peak numbers. We describe this analysis (subsection “MYC binding to chromatin appears saturated at certain sites”, second paragraph) in the Results section. We agree with the reviewers that most expressed genes are MYC target genes and added the sentence “In agreement with reports from other systems, we concluded that the promoters of the majority of all expressed genes are bound by MYC.” We reduced the suggested exactness when mentioning the number of MYC bound promoters, first (in the aforementioned paragraph).

*5) There are several issues related to the model. The authors only model the influence of WDR in concert with E-boxes; what is the level of binding and the influence of MYC at WDR^+^, E-box^-^ promoters? This class should be considered, too. What is the extent of MYC binding, expression levels and functional GSEA for this class of promoters?*

In response to this comment, we included the group of “WDR5-bound DNA sites without canonical E-boxes” as a fourth class in our model. The reviewers are correct in expecting that this is an interesting group, as binding to those sites is predicted to be high at endogenous MYC concentrations in U2OS cells (see new Figure 3; description in subsection “Binding to DNA and to WDR5 accounts for high promoter affinity”, third paragraph). Subsequently, we characterized these genes in terms of MYC binding status, gene ontology and expression status and present the results in the new Figure 4—figure supplement 2. As one might expect, MYC binding and gene expression of WDR5^+^CACGTG^-^-genes is higher than WDR5^-^CACGTG^-^-, but lower than WDR5^+^CACGTG^+^- genes (description in the aforementioned paragraph).

*6) The model assumes equilibrium (or at least steady state) binding. While this is often a reasonable assumption, at promoters, in vivo, there are many energy-dependent and kinetic steps, that might skew that apparent on and off rates. For example, the authors' own previous work showed that MYC degradation during the transcription-cycle might be required to increase expression. Such degradation will increase the apparent MYC off-rate, artificially depressing apparent affinity of MYC for the promoter. The kinetic limitations of the model should be discussed in the text or in the supplemental material.*

As the reviewers point out, binding of MYC to promoters can be affected by several regulated processes that may change the kinetic rates of association (“on-rate”) or of dissociation (“off-rate”). To our knowledge, experimental data on the kinetics of “on-rate” and “off-rate” constants or the respective kinetic rate laws of MYC is not yet available. Moreover, ChIP-sequencing experiments are based on the general notion that the number of DNA-bound molecules (tags) presumably does not change (at least) for the time frame of conducting the experiment. We therefore used the steady state assumption for calculating apparent dissociation constants, which imply ratios on the “on rates” and “off rates”. We mention this limitation in the main text (subsection 2 Binding to DNA and to WDR5 accounts for high promoter affinity”, first paragraph) and refer to appendix 1 for further discussion of this assumption. In order to check for the robustness of the model with respect to the specific values of these dissociation constants, we varied them over a broad range (see Figure 3) and discussed their impact on our conclusions (appendix 2).

*7) Some discussion of expression levels for the various functional classes of MYC-target promoters is warranted. It has previously been suggested that MYC binding is related to output levels. It may be that some cellular processes (such as translation) demand high output, and that it is the high level of transcription at these genes that licenses MYC binding, that in turn stimulates more transcription.*

Our analysis of MYC binding at endogenous concentrations clearly – and in agreement with the reviewers’ standpoint – demonstrated that target genes are characterized by high expression (Figure 4) and open chromatin structure (Figure 4). Accordingly, high affinity targets like ribosomal protein genes (Figure 2) are known to be among the highest transcribed genes. To make this correlation and the role of existing gene expression as a licensing factor for MYC binding most obvious for the reader, we introduced new statements like “These genes are thought to comprise a core signature of highly expressed MYC target genes (Ji et al., 2011).”, “Since target gene expression, H3K4me3 and the occurrence of E-box sequences stratify high affinity binding sites[…]”, “We propose that the dependence of MYC/MAX chromatin binding on interactions with proteins that are themselves influenced by existing gene expression patterns at different physiological states of cells, […]” and “This is likely to be true for strongly transcribed genes as many ribosomal protein genes […]” at various sections in the manuscript.

*Accompanying Tables of genes in groups identified in the Figures should be provided. (e.g. ribosome, extracellular matrix, solute transporter, etc.).*

Gene sets used in the study are now provided in [Supplementary-material SD2-data].

Further, the authors should provide an analysis of selected metabolic pathways, such as glycolysis and glutaminolysis, that may be regulated in U2OS cells.

We analyzed the EC_50_ values of selected metabolic pathways and included these analyses in Figure 5. Indeed, we observed elevated EC_50_ values for glycolysis, lipid- and purine metabolism (Figure 5, purple boxplots). The data are described in the last paragraph of the subsection “Binding affinity determines MYC-mediated transcriptional responses at different MYC concentrations” and discussed in the sixth paragraph of the Discussion.